# Diverse inhibitory projections from the cerebellar interposed nucleus

**Elena N Judd, Samantha M Lewis, Abigail L Person\***

Department of Physiology and Biophysics, University of Colorado School of Medicine, Anschutz Medical Campus, Aurora, United States

**Abstract** The cerebellum consists of parallel circuit modules that contribute to diverse behaviors, spanning motor to cognitive. Recent work employing cell-type-specific tracing has identified circumscribed output channels of the cerebellar nuclei (CbN) that could confer tight functional specificity. These studies have largely focused on excitatory projections of the CbN, however, leaving open the question of whether inhibitory neurons also constitute multiple output modules. We mapped output and input patterns to intersectionally restricted cell types of the interposed and adjacent interstitial nuclei in mice. In contrast to the widespread assumption of primarily excitatory outputs and restricted inferior olive-targeting inhibitory output, we found that inhibitory neurons from this region ramified widely within the brainstem, targeting both motor- and sensory-related nuclei, distinct from excitatory output targets. Despite differences in output targeting, monosynaptic rabies tracing revealed largely shared afferents to both cell classes. We discuss the potential novel functional roles for inhibitory outputs in the context of cerebellar theory.

## Introduction

The cerebellum plays a critical role in refining motor control through learning. The cerebellar nuclei (CbN), which constitute the major outputs of the cerebellum, are proposed to relay predictive computations of the cerebellar cortex and store well-learned patterns, placing them in a central position to implement cerebellar control (*Eccles and Szentágothai, 1967Ohyama et al., 2003*; *Chan-Palay, 1977*). The CbN are a collection of nuclei that house diverse neuronal subtypes that differ in their targets. Recent studies have greatly expanded our understanding of this diversity, using approaches such as genomic profiling and projection specific tracing (*Bagnall et al., 2009*; *Low et al., 2018*; *Fujita et al., 2020*; *Kebschull et al., 2020*; *Uusisaari et al., 2007*; *Uusisaari and Knöpfel, 2010*; *Uusisaari and Knöpfel, 2011*; *Husson et al., 2014*; *Ankri et al., 2015*; *Canto et al., 2016*). Through these studies, we know that multiple diverse output channels intermingle (*Fujita et al., 2020*; *Low et al., 2018*; *Sathyamurthy et al., 2020*), widespread collateralization is common, and genetic diversity of excitatory projection neurons varies systematically along the mediolateral extent of the CbN which encompasses the medial (fastigial), interposed, lateral (dentate), interstitial, and vestibular nuclei (*Kebschull et al., 2020*).

The mouse cerebellar interposed nucleus has received recent attention at the anatomical and functional levels with studies identifying specific projection patterns and functional roles for neuronal subtypes within the structure. Interposed excitatory neurons project to a variety of motor-related spinal cord and brainstem targets, as well as collateralize to motor thalamus (*Low et al., 2018*; *Sathyamurthy et al., 2020*; *Kebschull et al., 2020*). Ablation of a subset of anterior interposed (IntA) glutamatergic cells that express Urocortin3, for example, disrupts accurate limb positioning and timing during a reach to grasp task and locomotion (*Low et al., 2018*). Chemogenetic silencing of excitatory neurons that project ipsilaterally to the cervical spinal cord also impaired reach success in mice (*Sathyamurthy et al., 2020*). Moreover, closed-loop manipulation of IntA disrupts reach endpoint in real time (*Becker and Person, 2019*). The interposed nucleus also mediates conditioned eyelid responses, sculpts reach

---

**\*For correspondence:**
abigail.person@cuanschutz.edu

**Competing interest:** The authors declare that no competing interests exist.

and gait kinematics, and is responsive to tactile stimulation (*Darmohray et al., 2019*; *Ten Brinke et al., 2017*; *Rowland and Jaeger, 2005*). How anatomical organization of the structure confers such functions is an open question.

Functional consequences of cell-type-specific manipulations have not been limited to excitatory neurons. Ablation of inhibitory nucleo-olivary cells demarcated with Sox14 expression also resulted in motor coordination deficits (*Prekop et al., 2018*). These cells were traced from the lateral nucleus and suggested to project solely to the inferior olive (IO), consistent with conclusions from experiments using dual labeling methods (*Ruigrok and Teune, 2014*). Nevertheless, older reports of inhibitory projections from the CbN that target regions other than the IO raise the question of whether inhibitory outputs might also play a role in regulating brainstem nuclei outside the olivocerebellar system. Combined immunostaining with horseradish peroxidase tracing from the basilar pontine nuclei (i.e., pontine grey [PG]) in rats and cats showed GABA immunopositive neurons in the lateral nucleus (*Aas and Brodal, 1989*; *Border et al., 1986*), although the literature is inconsistent (*Schwarz and Schmitz, 1997*). Glycinergic output projections from the medial nucleus (fastigial) inhibitory output population include large glycinergic neurons that project to ipsilateral brainstem targets outside the IO (*Bagnall et al., 2009*), unlike its Gad2-expressing neurons which exclusively target the IO (*Fujita et al., 2020*). In aggregate, these various observations indicate that better understanding of whether the interposed nucleus houses inhibitory output neurons that project to targets outside IO is needed.

Here, we use a range of viral tracing methods to isolate and map projections from and to inhibitory and excitatory neurons of the intermediate cerebellar nuclear groups, defined through intersectional labeling methods using single or multiple recombinases coupled with pathway-specific labeling (*Fenno et al., 2014*). This method permitted analysis of collateralization more specific than traditional dual-retrograde labeling strategies since it leverages genetic specification and projection specificity and permits entire axonal fields to be traced. We elucidate the projection 'fingerprints' of genetic- and projection-defined cell groups. Surprisingly, we observed widespread inhibitory outputs, comprised at least in part of putative collaterals of some IO-projecting neurons, that target both ipsilateral and contralateral brainstem and midbrain structures. Monosynaptic rabies transsynaptic tracing (*Kim et al., 2016*; *Wickersham et al., 2010*) restricted to excitatory premotor neuron populations through the selective expression of Cre recombinase under the *Slc17a6* (Vglut2) promoter (*Gong et al., 2007*) and inhibitory neurons through Cre expression controlled under the *Slc32a1* (Vgat) promoter revealed reproducible patterns of presynaptic inputs largely shared across cell types. Taken together, these experiments provide new insight into input/output organization of the intermediate cerebellum, suggest potential functional diversity of parallel channels, and provide anatomical targets for functional studies aimed at evaluating these putative roles.

## Results

### Anterograde tracing of Int-Vgat neurons

To determine projection patterns of inhibitory neurons of the interposed nucleus, we stereotaxically injected AAV2-EF1a-DIO-YFP into Vgat-Cre transgenic mice, 'Int-Vgat,' (N=5, *Figure 1A*). We mapped and scored the extent and density of terminal varicosities on a 4-point scale and recorded injection sites, plotted for all experiments (*Figure 1—figure supplement 1*; see Materials and methods, Projection quantification).

As expected, injections labeled neurons that densely innervated the contralateral dorsal accessory IO (*Figure 1D, E and G*), with less dense but consistent innervation of ipsilateral IO (*Ruigrok and Voogd, 1990*; *Balaban and Beryozkin, 1994*; *Fredette and Mugnaini, 1991*; *Prekop et al., 2018Ruigrok and Voogd, 1990*; *Ruigrok and Voogd, 2000 Van der Want et al., 1989*). Surprisingly these injections also consistently labeled terminal fields outside IO, within the brainstem, even when injection sites were completely restricted to the anterior interposed nucleus. Viral expression of Int-Vgat neurons labeled axonal varicosities which were immunopositive for antibodies against Gad65/67, but never Vglut2, consistent with a GABAergic phenotype for these projections (*Figure 1E*, *Figure 1—figure supplement 2*; analyzed in the IO, spinal trigeminal nucleus, interpolar (SPVi), PG, red nucleus (RN), and vestibular nuclei). In situ hybridization (ISH) revealed that 98 % of virally labeled cells co-expressed the Vgat gene *Slc32a1* (230/234 cells from two mice), while 4/234 cells overlapped the Vglut2 gene *Slc17a6* (*Figure 1F*, *Figure 1—figure supplement 3*, *Supplementary file*

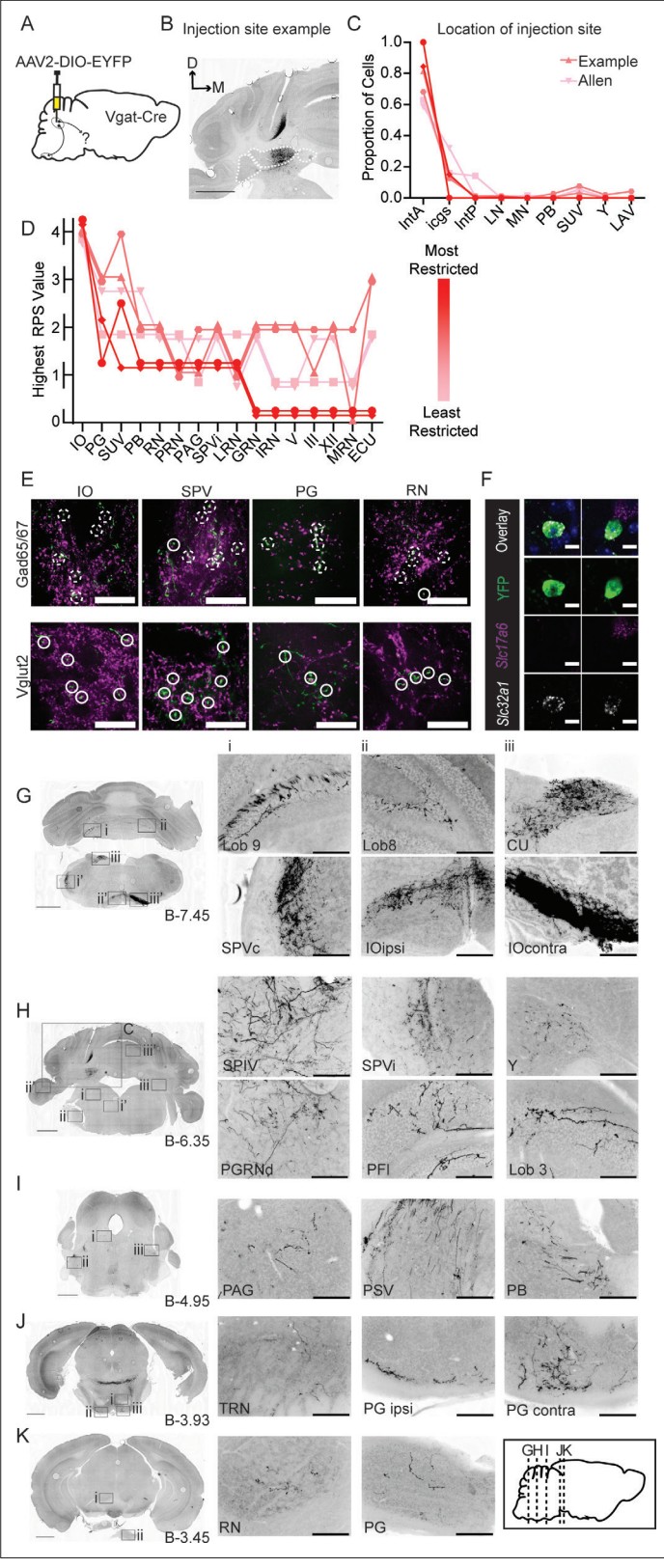

**Figure 1.** Anterograde tracing of Int-Vgat neurons. (**A**) Schematic of injection scheme. (**B**) Example injection site of AAV2-EF1a-DIO-eYFP. The three main CbN are outlined in white (lateral nucleus (LN), interposed (IN), and medial nucleus (MN) from left to right). Images oriented so that the dorsal-ventral axis runs up/down and the medial/lateral axis runs right/left; right of midline is contralateral. (**C**) Location of labeled cells by injection into

*Figure 1 continued on next page*

*Figure 1 continued*

Int of Vgat-Cre mice. Specimens are color-coded by the proportion of cells labeled in anterior interposed (IntA) where the highest proportion corresponds to darkest color. (**D**) Mapping of terminal fields based on restriction of injection site to IntA. The highest unilateral relative projection strength (RPS) in each region is plotted for all specimens included in analysis. The values are always assigned as integers but are offset here so overlap can be better appreciated. (**E**) YFP-positive terminals (green) in inferior olive (IO), spinal trigeminal nuclei (SPVc), pontine grey (PG), and red nucleus (RN) are stained for antibodies against Gad65/67 (top) and Vglut2 (bottom; magenta). Dashed circles indicate colocalized terminals while solid lines indicate a lack of colocalization observed in the two channels. Scale bars=20 µm. (**F**) Example cells from in situ hybridization showing clear overlap with an mRNA probe against *Slc32a1* (Vgat) and no overlap with an mRNA probe against *Slc17a6* (Vglut2). Scale bars=10 µm. (**G**) Projection targets in caudal cerebellum and brainstem (B-7.45). Boxes expanded in (**i–iii**) (top) or (**i–iii′**) (bottom). (**H**) Projection targets within the intermediate cerebellum (B-6.35). Injection site depicted in (**C**). (**I**) Projection targets within rostral brainstem (B-4.95). (**J**) Projection targets in the caudal midbrain (B-3.93). (**K**) Projection targets to the rostral midbrain (B-3.93). Scale bars (**C, G–K**) =1 mm and (**i–iii**) 200 µm. The inset (black border) depicts the location of coronal sections shown in (**G–K**) along a parasagittal axis. Cuneate nucleus (CU), gigantocellular reticular nucleus (GRN), hypoglossal nucleus (XII), intermediate reticular nucleus (IRN), interstitial cell groups (icgs), lateral reticular nucleus (LRN), lateral vestibular nucleus (LAV), midbrain reticular nucleus (MRN), motor nucleus of the trigeminal (V), nucleus prepositus (PRP), Nucleus Y (Y), oculomotor nucleus (III), parabrachial (PB), paraflocculus (PFl), paragigantocellualr reticular nucleus (PGRN), periaqueductal grey (PAG), principle sensory nucleus of the trigeminal (PSV), pontine reticular nucleus (PRN), posterior interposed (IntP), spinal trigeminal nucleus, caudal/interpolar subdivision (SPVc/i), spinal vestibular nucleus (SPIV), superior vestibular nucleus (SUV), and tegmental reticular nucleus (TRN).

The online version of this article includes the following figure supplement(s) for figure 1:

**Figure supplement 1.** Example of semiquantitative scoring method of terminal field extent and density in Ntsr1 and Vgat-Cre mice.

**Figure supplement 2.** Immunoreactivity of Vgat-Cre terminal varicosities.

**Figure supplement 3.** In situ hybridization (ISH) methods and analysis.

**Figure supplement 4.** Gad1-Cre localized to mulitple cellular phenotypes in Int.

**Figure supplement 5.** Example projections from an IntA restricted Int-Vgat specimen.

---

*2*). A Gad1-Cre driver line (*Higo et al., 2009*) was tested but not used owing to non-specific label (*Figure 1—figure supplement 4*; see Materials and methods; *Supplementary file 2*).

Most Int-Vgat injections included both interposed and interstitial cell groups slightly ventral to the interposed nucleus, plotted in *Figure 1B*, color-coded for the proportion of the injection site contained within IntA. Although injection site spillover into interstitial cell groups (*Sugihara and Shinoda, 2007*) was common, injection site spillover into the main vestibular groups ventral to the fourth ventricle was minimal to absent. Following these injections, terminal label within the brainstem was extensive, and invariably also included beaded varicosities within the cerebellar cortex characteristic of the inhibitory nucleocortical pathway (*Ankri et al., 2015*). Modestly dense but spatially extensive terminal fields ramified in the posterior medulla along the anterior-posterior axis (*Figures 1 and 2D*); no retrogradely labeled neurons were observed. Among sensory brainstem structures, terminal fields ramified within the ipsilateral external cuneate nucleus (ECU), cuneate nucleus (CU), nucleus of the solitary tract (NTS), SPVi, especially the lateral edge, parabrachial nuclei (PB), principal sensory nuclei of the trigeminal (PSV), and all vestibular nuclei. Int-Vgat axons extended through the pontine reticular nuclei (PRN) to innervate the tegmental reticular nuclei (TRN; commonly abbreviated NRTP) and the PG (i.e., basilar pontine nuclei; *Figure 1D and J*), which are themselves major sources of cerebellar mossy fibers. Int-Vgat neurons also innervated the medial magnocellular RN (*Figure 1K*) bilaterally. Rarely, Int-Vgat axons progressed to the caudal diencephalon, very sparsely targeting the ipsilateral zona incerta ZI in 2/6 mice (*Supplementary file 3*). Axonal varicosities were vanishingly sparse or non-existent within the spinal cord following Int-Vgat injections (data not shown).

Beaded nucleocortical fibers from Int-Vgat injections were reliably labeled if the injection site included interstitial cell groups (*Figures 1G, H and 2D*; *Ankri et al., 2015*). Int-Vgat neurons targeted all cerebellar lobules, even extending contralaterally.

Some targets noted were sensitive to injection site restriction (*Figure 1D*). However, labeling of varicosities outside IO was not attributable solely to injection site leakage outside Int. The smallest Int-Vgat injection, contained entirely within IntA, labeled fine caliber axons that ramified within the

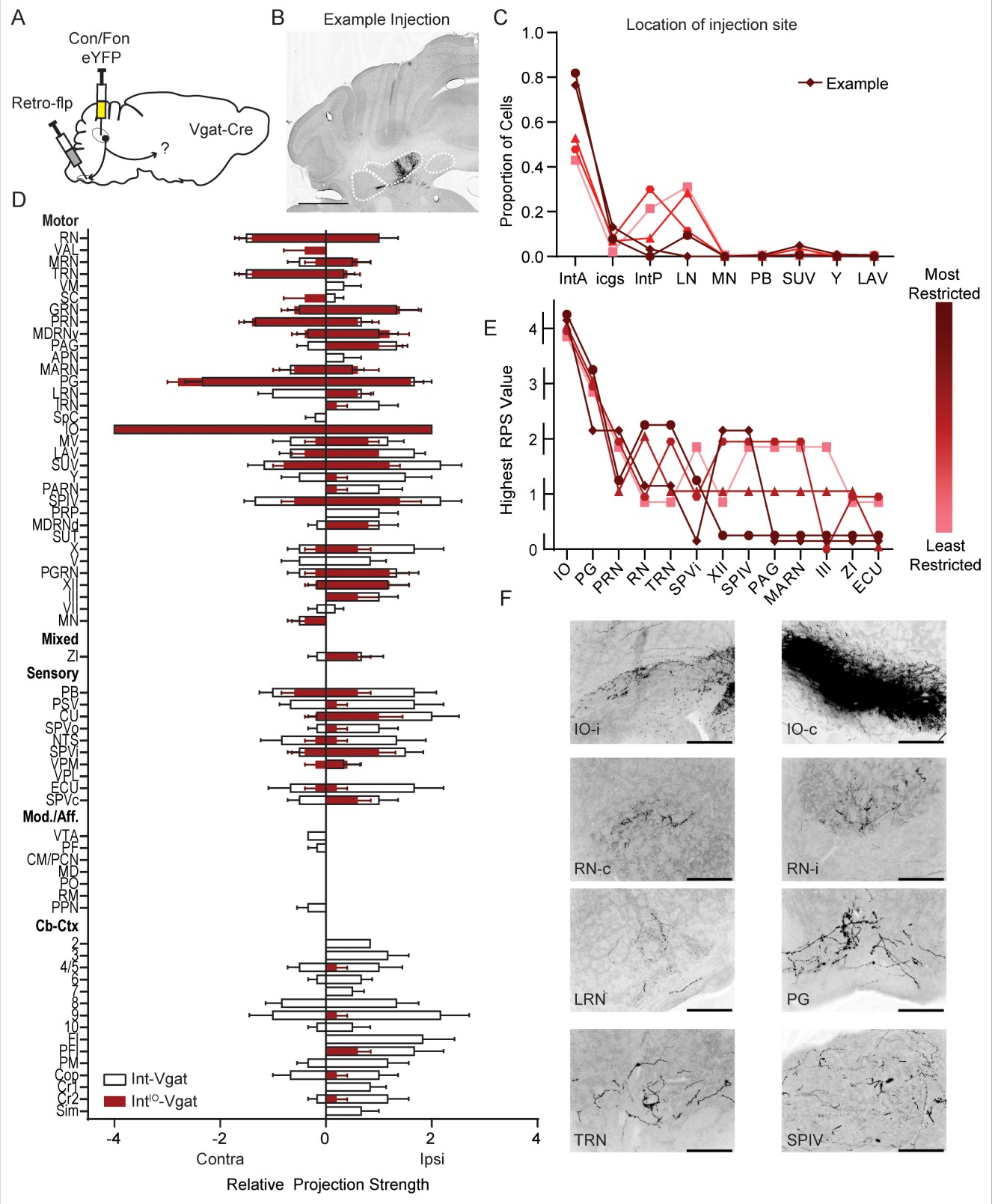

**Figure 2.** Intersectional labeling of IO-projecting Int-Vgat neurons (Int[IO]-Vgat) and comparison with Int-Vgat. (**A**) Schematic of experiment. (**B**) Example injection site of AAV8.hSyn.Con/Fon.hChR2.EYFP in a Vgat-Cre mouse. The three main CbN are outlined in white (lateral nucleus (LN), interposed (IN), and medial nucleus (MN) from left to right). Images oriented as in **Figure 1**. Scale bars=1 mm. (**C**) Location of labeled cells by injection of Retro-Flp to the contralateral inferior olive (IO) and Con/Fon-YFP into Int of Vgat-Cre mice. Specimens are color-coded by the proportion of cells labeled in anterior

*Figure 2 continued on next page*

*Figure 2 continued*

interposed (IntA) where the highest proportion corresponds to darkest color. (**D**) Graphical representation of average projection strength in all targeted regions for Int^IO-Vgat (n=5; maroon) and Int-Vgat (n=6; white) mice. See the list of abbreviations for complete listing. (**E**) Mapping of terminal fields based on restriction of injection site to IntA. The highest unilateral RPS in each region is plotted for all specimens included in analysis. The values are always assigned as integers but are offset here so overlap can be better appreciated. (**F**) Example terminal fields within the (IO) and red nucleus (RN) bilaterally, lateral reticular nucleus (LRN), pontine grey (PG), tegmental reticular nucleus (TRN), and spinal vestibular nucleus (SPIV). Scale bars=200 µm. External cuneate nucleus (ECU), hypoglossal nucleus (XII), interstitial cell groups (icgs), lateral vestibular nucleus (LAV), magnocellular reticular nucleus (MARN), Nucleus Y (Y), oculomotor nucleus (III), parabrachial (PB), periaqueductal grey (PAG), pontine reticular nucleus (PRN), posterior interposed (IntP), spinal trigeminal nuclei, interpolar (SPVi), spinal vestibular nucleus (SPIV), superior vestibular nucleus (SUV), and zona inserta (ZI).

The online version of this article includes the following figure supplement(s) for figure 2:

**Figure supplement 1.** Viral control injections.

ipsilateral superior and spinal vestibular nuclei (*Figure 1—figure supplement 5*). Labeled fibers coursed in the superior cerebellar peduncle, decussating at the level of the pontine nuclei (–4 mm Bregma). As they coursed ventrally, they produced numerous varicosities in the pontine nuclei, specifically the tegmental reticular nucleus and PG, before turning caudally, labeling dense terminals fields in the contralateral IO (DAO) and modestly dense fields in the ipsilateral IO. Very sparse varicosities were also noted in the parabrachial nucleus and magnocellular RN. Despite the presence of these terminal fields, no nucleocortical fibers were seen following the most restricted Int-Vgat injection, suggesting these may originate from interstitial cell groups and/or other CbN. To summarize, Int-Vgat injections labeled fibers that innervated numerous brainstem nuclei outside IO, even following highly restricted injections.

## Projection-specific Int-Vgat neuron tracing

The terminals observed in the brainstem and midbrain from Int-Vgat labeling suggested the existence of inhibitory channels from the intermediate cerebellum beyond those targeting the IO.

Next, to restrict label to genetic- and projection-specific Int neurons (*Fenno et al., 2014*), we used a two-recombinase-dependent reporter virus (AAV8-hsyn-Con/Fon-eYFP) injected into Int in conjunction with Flp recombinase retrogradely introduced via the contralateral IO with AAVretro-EF1a-Flp (*Figure 2A*; N=5). The fluorescent reporter will only express in the presence of both Cre and Flp recombinases. This Cre-on Flp-on approach, termed 'Con/Fon,' was used to isolate IO-projecting Int-Vgat neurons. Specificity was determined via injections in wild-type C57/Bl6 mice (N=2) and off-target injections in Cre mice (N=3), which did not yield YFP positive neurons in the CbN (*Figure 2—figure supplement 1*).

Int^IO-Vgat neurons had more restricted terminations than most direct Int-Vgat injections. Varicosities were consistently observed in dorsal PG, PRN, TRN, IO, and the vestibular complex. Less consistent and sparser label occurred in other brainstem nuclei (*Figure 2*). These data are consistent with either of two non-mutually exclusive possibilities: that at least some IO-projecting cells collateralize to a subset of targets relative to the constellation of regions targeted by all Int-Vgat neurons, typically excluding nucleocortical projections, modulatory/affective regions, and sensory nuclei; and/or these intersectional methods restrict the range of neurons infected owing to IO targeting, which restricts the other axonal fields labeled. We also note that this result does not preclude the existence of IO-only projecting neurons, which was not directly examined.

## Anterograde tracing from excitatory output neurons

To compare Int-Vgat projections more directly to excitatory outputs, we injected Int of Ntsr1-Cre mice with AAV1-CAG-flex-GFP (N=2) or AAV2-DIO-EF1a-eYFP (N=3) (*Figure 3*). Int-Ntsr1 terminal varicosities consistently colocalized with Vglut2 immunolabel, but never Vgat, consistent with a glutamatergic phenotype of Ntsr1 output neurons (*Figure 3E*, *Figure 3—figure supplement 1*), and somata overlapped predominantly with the glutamatergic marker *Slc17a6* (*Figure 3F*; *Figure 1—figure supplement 3*). Dense and consistent terminal varicosities labeled by Int-Ntsr1 neurons occurred in patches within the caudal medulla, midbrain, and thalamus, which are known targets of Vglut2-Cre and Ucn3-Cre neurons (*Figure 3D and G–K*, *Low et al., 2018*; *Kebschull et al., 2020*; *Sathyamurthy et al., 2020*). Varicosities filled the ipsilateral parvicellular reticular nucleus PARN (commonly abbreviated PCRt) which extended rostrally to blend into the spinal

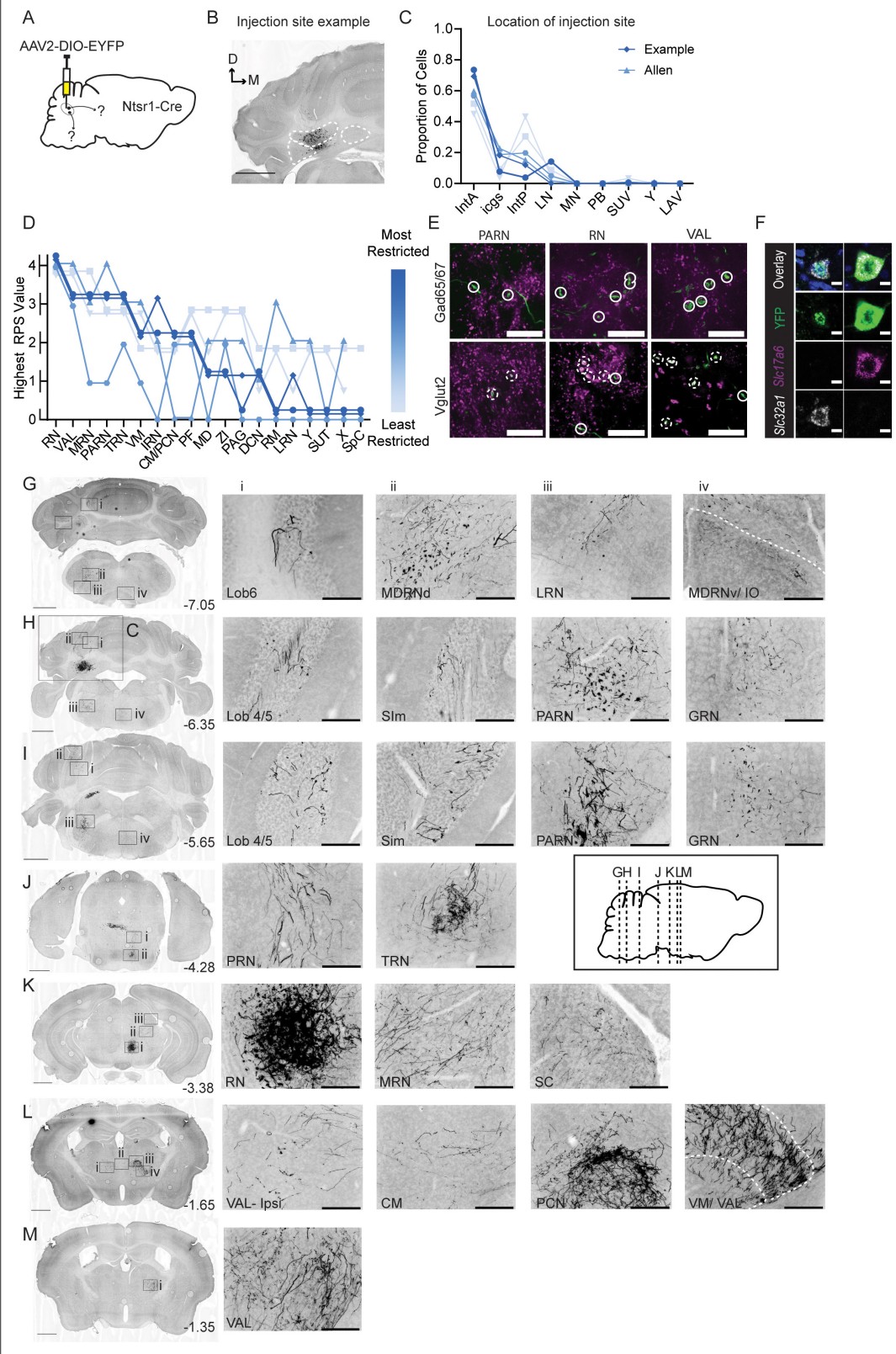

**Figure 3.** Anterograde tracing of Int-Ntsr1 neurons. (**A**) Schematic representation of injection scheme. (**B**) Example injection site of AAV2-EF1a-DIO-eYFP. The three main CbN are outlined in white (Lateral nucleus (LN), interposed (IN), and medial nucleus (MN) from left to right). Images oriented as in *Figure 1*. (**C**) Distribution of labeled cells by injection into Int of Ntsr1-Cre mice. Specimens are color-coded by the proportion of cells

*Figure 3 continued on next page*

*Figure 3 continued*

labeled in anterior interposed (IntA) where the highest proportion corresponds to darkest color. (**D**) Mapping of terminal fields based on restriction of injection site to IntA. The highest unilateral RPS in each region is plotted for all specimens included in analysis. The values are always assigned as integers but are offset here so overlap can be better appreciated. (**E**) YFP-positive terminals (green) in parvicellular reticular nucleus (PARN), red nucleus (RN), and ventral anterior-lateral complex of the thalamus (VAL) are stained for antibodies against Gad65/67 (top; magenta) and Vglut2 (bottom; magenta). Dashed circles indicate colocalized terminals while solid lines indicate a lack of colocalization observed in the two channels. Scale bars=20 µm. (**F**) Example cells from in situ hybridization showing overlap with both an mRNA probe against *Slc32a1* (Vgat) and *Slc17a6* (Vglut2). Scale bars=10 µm. (**G**) Projection targets in caudal cerebellum and brainstem (B-7.05). Boxes expanded in (**i–iv**). (**H**) Projection targets within the intermediate cerebellum (B-6.35). Injection site depicted in (**C**). (**I**) Projection targets within and ventral to the anterior cerebellum (B-5.65). (**J**) Projection targets to pontine nuclei (B-4.25). (**K**) Projection targets in the rostral midbrain (B-3.38). Note the dense terminals in RN. (**L**) Projection targets to the caudal thalamus (B-1.65). (**M**) Projection targets to the rostral thalamus (B-1.35). Scale bars (**C, G–M**) =1 mm and (**i–iv**) 200 µm. The inset (black border) depicts the location of coronal sections shown in (**G–M**) along a parasagittal axis. Centromedial nucleus of the thalamus (CM), cuneate nucleus (CU), gigantocellular reticular nucleus (GRN), inferior olive (IO), intermediate reticular nucleus (IRN), interstitial cell groups (icgs), lateral reticular nucleus (LRN), lateral vestibular nucleus (LAV), mediodorsal nucleus of the thalamus (MD), medullary reticular nucleus, dorsal/ventral subdivision (MDRNd/v), midbrain reticular nucleus (MRN), nucleus raphe magnus (RM), nucleus X (X), nucleus Y (Y), parabrachial (PB), paracentral nucleus of the thalamus (PCN), parafascicular nucleus (PF), periaqueductal grey (PAG), pontine reticular nucleus (PRN), posterior interposed (IntP), simplex lobule (Sim), superior colliculus (SC), superior vestibular nucleus (SUV), supratrigeminal nucleus (SUT), spinal cord (SpC), tegmental reticular nucleus (TRN), ventromedial nucleus (VM), and zona incerta (ZI).

The online version of this article includes the following figure supplement(s) for figure 3:

**Figure supplement 1.** Immunoreactivity of Ntsr1-Cre terminal varicosities.

**Figure supplement 2.** Int-Ntsr1 neurons (green) target tyrosine hydroxylase (TH; magenta) expressing neurons in the ventral tegmental area (VTA).

nucleus of the trigeminal (SPV), known forelimb control structures (*Esposito et al., 2014*), and ipsilateral terminals ramified in the motor nucleus of the trigeminal (V). Bilateral patches of terminals were seen in the lateral reticular nucleus (LRN) and all four subdivisions of the vestibular nuclei. At the level of the decussation of the superior cerebellar peduncle, axons turned ventrally and produced dense Vglut2-positive varicosities in the TRN (commonly abbreviated NRTP) and sparsely in PG (*Cicirata et al., 2005*; *Schwarz and Schmitz, 1997*). Axons also ramified within the magnocellular RN and the deep layers of the superior colliculus (SC). Diencephalic projections were densely targeted to thalamic nuclei and more sparsely targeted to ZI. All specimens exhibited dense terminal fields in the ventromedial (VM) and anterior ventrolateral (VAL) nuclei of the thalamus (*Teune et al., 2000*; *Aumann et al., 1994*; *Houck and Person, 2015*; *Kalil, 1981*; *Low et al., 2018*; *Stanton, 1980*). Additionally, we observed terminals in intralaminar thalamic structures including centromedial (CM), paracentral (PCN), mediodorsal (MD), parafascicular (PF), ventral posterior (VP), and posterior (PO) nuclei (*Teune et al., 2000*; *Chen et al., 2014*; *Dumas et al., 2019*). Int-Ntsr1 neurons formed nucleocortical mossy fibers in multiple lobules across the cortex (*Figure 3G–I*; *Gao et al., 2016*; *Houck and Person, 2015*; *Tolbert et al., 1978*; *Low et al., 2018*; *Sathyamurthy et al., 2020*).

Beyond the major targets described above, Int-Ntsr1 projected sparsely to a variety of other regions. In three out of five animals, we observed a small patch of terminals within the contralateral dorsal subnucleus of IO that were positive for Vglut2 (*Figure 3G*, *Figure 3—figure supplement 1*). Near the dense terminal field within the contralateral RNm, fine caliber axons bearing varicosities spilled over into the ventral tegmental area, VTA (*Figure 3—figure supplement 2*; *Carta et al., 2019*; *Teune et al., 2000*) and extended dorsally through the contralateral midbrain/mesencephalic reticular nucleus (MRN; *Ferreira-Pinto et al., 2021*) to innervate the caudal anterior pretectal nucleus (APN) anterior ventrolateral periaqueductal grey (PAG) (*Vaaga et al., 2020*; *Sugimoto et al., 1982*; *Gayer and Faull, 1988*; *Low et al., 2018*; *Teune et al., 2000*). To summarize, Int-Ntsr1 neurons targeted regions well known to receive excitatory input from the interposed nucleus, as well as a previously unappreciated sparse Vglut2+ afferent to the IO.

## Projection-specific Int-Ntsr1 neuron tracing

We next used the Con/Fon intersectional approach described above to restrict labeling to RN-projecting Ntsr1-Cre neurons (Int$^{RN}$-Ntsr1, *Figure 4*; N=4), asking whether projection-specific labeling recapitulated data from direct label of Int-Ntsr1 cells, as would be expected if RN projecting neurons collateralize to other targets. The projection pattern of Int$^{RN}$-Ntsr1 was almost identical to the pattern observed in Int-Ntsr1 injections, with a few notable exceptions. Namely, only Int-Ntsr1 neurons projected to lobule 8, anterior pretectal nucleus (APN), IO, and pedunculopontine nuclei (PPN). Terminal fields in the contralateral thalamus, especially VAL, VM, and CM/ PCN as well as layers 7/8 of the contralateral cervical spinal cord (2/3 specimens with spinal cords available) support the observation in *Sathyamurthy et al., 2020* that contralaterally projecting cerebellospinal neurons collateralize to both RN and thalamus. We conclude that it is likely that Int-Ntsr1 neurons reliably project to RN and collateralize to a restricted collection of other targets, although these data do not distinguish between broad versus restricted collateralization of Int$^{RN}$-Ntsr1 neurons.

## Projections of IntA$^{RN}$ neurons traced with AAVretro-Cre

As described above, we noted that both Int-Vgat and Int-Ntsr1 labeled varicosities within RN. This presented a target that we could exploit to test whether Int neurons collateralize to both RN and IO independent of genetic Cre label. We retrogradely expressed Cre in RN-projecting neurons, injecting AAV2retro-Cre into RN and a flexed reporter virus into Int (AAV1-CAG-flex-GFP/ RFP) of wild-type C57/Bl6 mice (*Figure 4—figure supplement 1*; N=4). Following these injections, we observed label in both IO and RN contralateral to the Int injection (*Figure 4—figure supplement 1*; *Supplementary files 1 and 3*). We also observed terminals in other locations consistently targeted by either Int-Ntsr1 (MRN, VAL, VPM, VM, PF, MD, PO, SC, and ZI) or Int-Vgat (Lob 9, IO, lateral SPV, ipsilateral PRN, and ECU). Following these injections, terminal varicosities in IO, and subsets in TRN and PG expressed Gad65/67 while varicosities in SPV, RN, PG, VAL, and TRN were positive for Vglut2 (*Figure 4—figure supplement 2*; N=2). We conclude that retrograde uptake of Cre from synaptic terminals in RN results in reporter expression of both glutamatergic and GABAergic neurons in Int that both project to RN.

## Comparison of projection patterns across labeling methods

Across the distinct labeling methods, we observed a variety of notable patterns that differentiated them. First, Int cell sizes differed by Cre driver lines. We measured the cross-sectional area and elliptical diameter of somata of virally labeled cells. Int-Vgat neurons tended to be small with tortured dendrites (*Figure 5A–C*; 14.4±0.5 µm diameter, 95% confidence interval [CI]=[12.99, 15.85]; 109.3±7.8 µm$^2$ area, 95% CI=[87.54, 131.1]; N=5 mice; n=316 neurons). By contrast, Int-Ntsr1 neurons were characteristically large with smooth dendrites (*Figure 5A–C*; 22.2±0.8 µm diameter, 95% CI=[20.03, 24.37]; 224.7± 13.6 µm$^2$ area, 95% CI=[186.9, 262.5]; N=5 mice; n=229 neurons). Similarly, Int$^{IO}$-Vgat neurons were small (*Figure 5B–C*; 14.5±0.2 µm diameter, 95% CI=[13.92, 15.07]; 103.4±2.2 µm$^2$ area, 95% CI=[97.42, 109.5]; N=5 mice; n=404 neurons) and Int$^{RN}$-Ntsr1 neurons larger (*Figure 5B–C*; 22.1±1.0 µm diameter, 95% CI=[19.02, 25.27]; 238.2±18.5 µm$^2$ area, 95% CI=[179.4, 297.1]; N=4 mice; n=125 neurons). We compared these groups statistically and found that the Int-Vgat and Int$^{IO}$-Vgat cells were not significantly different from one another but were significantly smaller than Int-Ntsr1 and Int$^{RN}$-Ntsr1 neurons (one-way analysis of variance [ANOVA]; Tukey's multiple comparison test; $F_{(3,15)}$=46.99, p<0.0001 for all cross-genotype comparisons of means across specimens; p>0.99 for all within genotype comparisons of means across specimens).

Second, we noted that many targets were distinct between genotypes and projection classes. We classified extracerebellar target regions as motor, sensory, and modulatory, based in part on groupings of the Allen Brain Atlas (see Materials and methods). Notably, aggregate projection strength analyses indicated that on average, Int-Vgat neurons targeted sensory structures more densely than Int-Ntsr1 neurons (*Figure 5D*; t(7.6)=4.9, p=0.001, unpaired Welch's t-test). By contrast, we observed significantly stronger innervation of modulatory regions by Int-Ntsr1 than Int-Vgat (*Figure 5D*; p=0.0002, t(5.4) = 8.6, unpaired Welch's t-test). Additionally, Int-Ntsr1 projections showed a contralateral bias and Int-Vgat an ipsilateral bias, but these trends were not significant when accounting for false positive discovery rates (*Figure 5E and F*; t(6.9)=3.1, p=0.02, unpaired Welch's t-test).

Third, qualitative assessment showed that axons tended to ramify in distinct subdivisions within the subset of targets shared by Int-Vgat and Int-Ntsr1. For example, Int-Vgat neurons projected to more

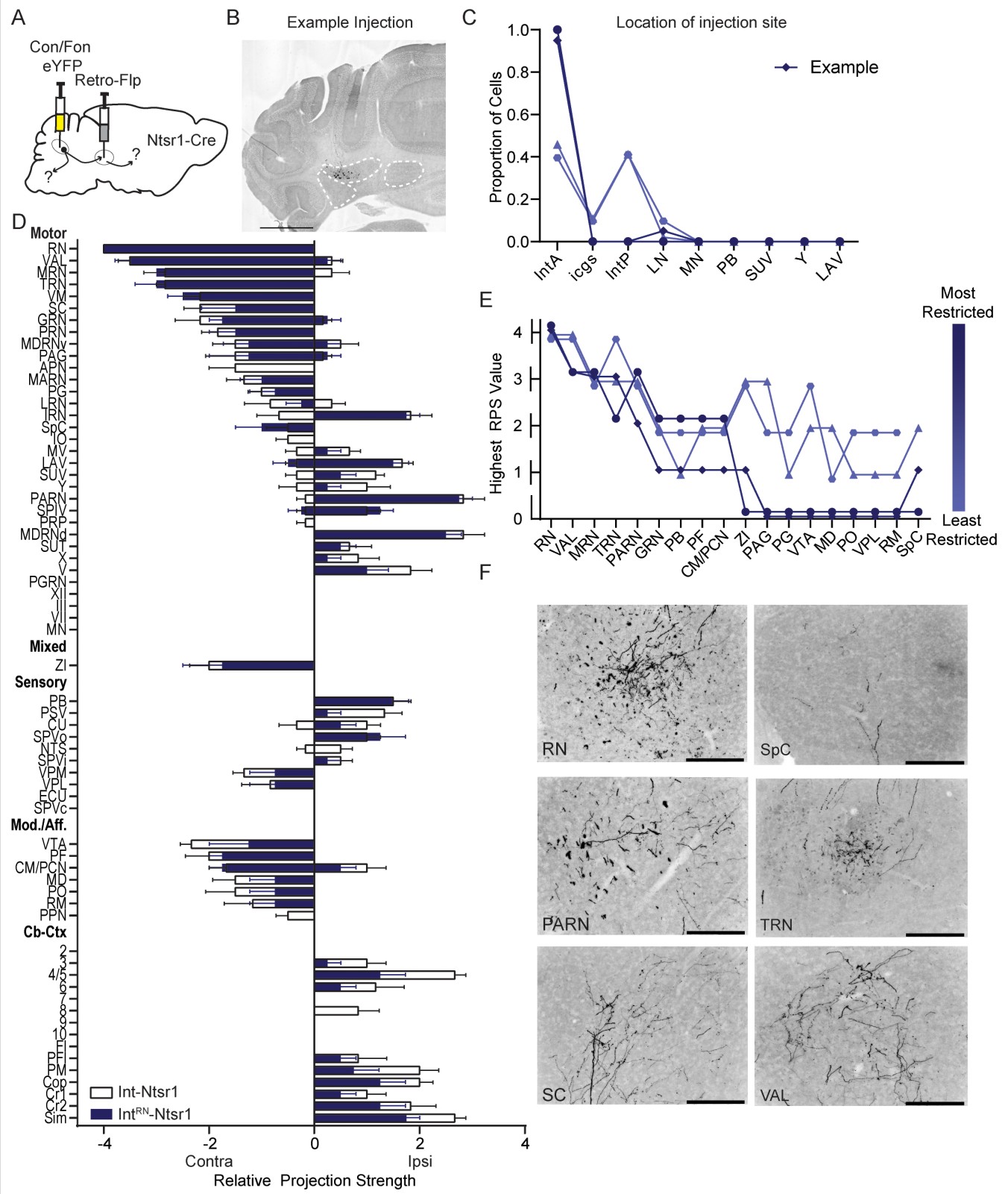

**Figure 4.** Intersectional labeling of RN-projecting Int-Ntsr1 neurons (Int^RN-Ntsr1). (**A**) Schematic of experiment. (**B**) Example injection site of AAV8. hSyn.Con/Fon.hChR2.EYFP in an Ntsr1-Cre mouse. The three main CbN are outlined in white (lateral nucleus (LN), interposed (IN), and medial nucleus (MN) from left to right). Images oriented so right of midline is contralateral. Scale bars=1 mm. (**C**) Location of labeled cells by injection of Retro-Flp to the contralateral red nucleus (RN) and Con/Fon-YFP into Int of Ntsr1-Cre mice. Specimens are color-coded by the proportion of cells labeled in

*Figure 4 continued on next page*

*Figure 4 continued*

anterior interposed (IntA) where the highest proportion corresponds to darkest color. (**D**) Graphical representation of average projection strength in all targeted regions for Int$^{RN}$-Ntsr1 (n=4; navy) and Int-Ntsr1 (n=6; white) mice. See the list of abbreviations for complete listing. (**E**) Mapping of terminal fields based on restriction of injection site to IntA. The highest unilateral RPS in each region is plotted for all specimens included in analysis. The values are always assigned as integers but are offset here so overlap can be better appreciated. (**F**) Example terminal fields within the red nucleus (RN), spinal cord (SpC), parvicellular reticular nucleus (PARN), tegmental reticular nucleus (TRN), superior colliculus (SC), and ventral anterior-lateral complex of the thalamus (VAL). Scale bars=200 μm. Centromedial nucleus of the thalamus (CM), gigantocellular reticular nucleus (GRN), inferior olive (IO), intermediate reticular nucleus (IRN), interstitial cell groups (icgs), lateral reticular nucleus (LRN), lateral vestibular nucleus (LAV), mediodorsal nucleus of the thalamus (MD), midbrain reticular nucleus (MRN), motor nucleus of the trigeminal (**V**), nucleus raphe magnus (RM), nucleus X (X), nucleus Y (Y), parabrachial (PB), paracentral nucleus of the thalamus (PCN), parafascicular nucleus (PF), periaqueductal grey (PAG), pontine grey (PG), pontine reticular nucleus (PRN), posterior complex of the thalamus (PO), posterior interposed (IntP), superior vestibular nucleus (SUV), supratrigeminal nucleus (SUT), tegmental reticular nucleus (TRN), ventral tegmental area (VTA), ventromedial nucleus (VM), ventral posterolateral nucleus of the thalamus (VPL), and zona incerta (ZI).

The online version of this article includes the following figure supplement(s) for figure 4:

**Figure supplement 1.** Labeling of RN-projecting Int neurons using viral Cre delivery.

**Figure supplement 2.** Immunoreactivity of Int-RetroCre$^{RN}$ terminal varicosities.

lateral regions of the caudal spinal nucleus of the trigeminal (SPVc), and to more lateral and anterior divisions of the principle sensory nucleus of the trigeminal (PSV). Int-Ntsr1 projected to the medial edge of SPVc near the border with MDRNd/ PARN and to PSV near the border of the trigeminal (V). While both Int-Vgat and Int-Ntsr1 projected to the vestibular nuclei, Int-Vgat projections ramified more caudally in the spinal and medial nuclei than Int-Ntsr1. Int-Vgat projections to the SC were absent. We also noted striking distinctions in the midbrain, where fibers from the two genotypes coursed in distinct locations. After decussating, Int-Vgat axons coursed farther lateral before turning ventrally toward the pontine nuclei. By contrast, Int-Ntsr1 axons turned ventrally at more medial levels after decussation, near the medial tracts through the pontine reticular nucleus (*Figure 5G*). Because injection sites did not differ qualitatively across injection types, we interpret these distinctions to reflect targeting differences across cell classes.

Finally, as has been noted in previous studies, nucleocortical fiber morphology differs between excitatory and inhibitory neurons (*Ankri et al., 2015*; *Houck and Person, 2015*; *Gao et al., 2016*; *Batini et al., 1992*). Int-Vgat injections labeled beaded varicosities devoid of mossy fiber morphological specializations (*Figure 5H*, top panels). Int-Ntsr1 labeled terminals with typical mossy fiber endings, large excrescences with fine filopodial extensions, and these predominantly targeted more intermediate lobules. Additionally, terminals in RN from Int-Vgat were very fine caliber while those from Int-Ntsr1 had thicker axons (*Figure 5H*, bottom panels). While these observations are qualitative in nature, they align with the small cellular morphology of Int-Vgat neurons relative to Int-Ntsr1 neurons.

## Cell-type-specific input tracing using monosynaptic rabies virus

Having mapped pathways from diverse cell types of the intermediate CbN, we next investigated afferents to these cells (*Figure 6A*). As described above, ISH in Vgat-Cre mice validated Vgat somatic expression in YFP labeled cells within the interposed nucleus, thus these mice were used for input tracing to inhibitory neurons (n=3). However, although output tracing from Int-Ntsr1 was validated with immunolabeling of terminals varicosities against Vglut2, ISH analysis of Ntsr1-Cre revealed 89 % of YFP-positive cell bodies expressed *Slc17a6* (Vglut2) probes (*Figure 3F*, *Figure 1—figure supplement 3*; 119/132 cells in two mice) but some labeled cells, possibly interneurons, expressed *Slc32a1* (Vgat) (15/132 cells in two mice). Thus, to ensure input mapping specific to excitatory neurons, we tested mRNA probe specificity of Vglut2-Cre mice (*Figure 6B*, *Figure 1—figure supplement 3*, *Supplementary file 2*): 178/179 YFP expressing cells (99%) expressed *Slc17a6* (Vglut2) mRNA and 3/179 expressed *Slc32a1* (Vgat) probes. Therefore, Vglut2-Cre (n=3) mice were used to isolate inputs to glutamatergic Int populations. These mice were used in conjunction with modified rabies (EnvA-ΔG-Rabies-GFP/mCherry) and Cre-dependent receptor and transcomplementation helper viruses (*Figure 6A*; see Materials and methods; *Kim et al., 2016*; *Wall et al., 2010*; *Watabe-Uchida et al., 2012*; *Wickersham et al., 2007*; *Wickersham et al., 2010*). Direct rabies virus infection was limited to cells that expressed the receptor, TVA; transsynaptic jump was restricted by complementation of optimized rabies glycoprotein (oG). In a subset of experiments, oG was restricted to TVA-expressing

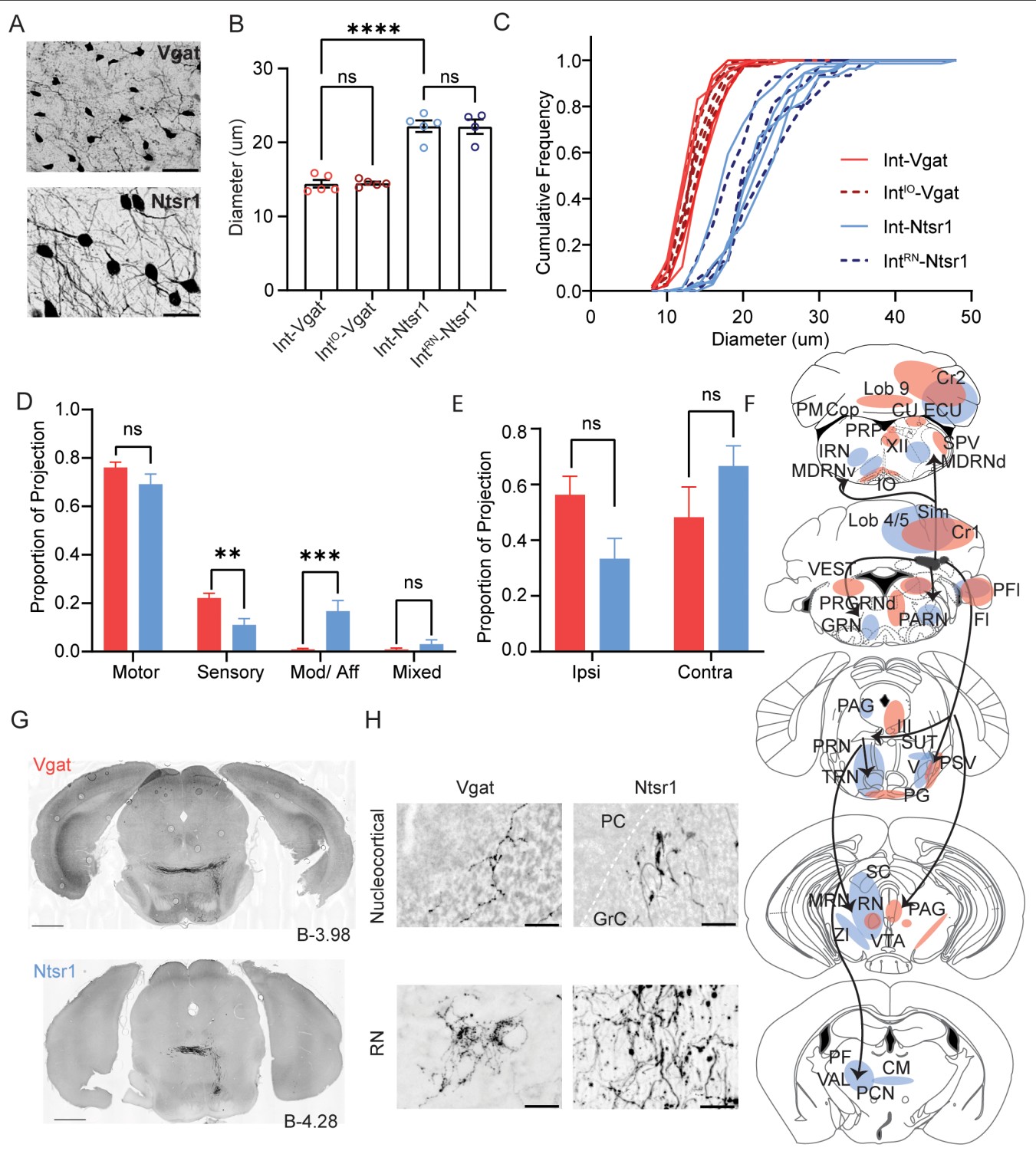

**Figure 5.** Comparison Int-Vgat and Int-Ntsr1 cell sizes and projection patterns. (**A**) Example YFP+ cells in a Vgat-Cre (top) and Ntsr1-Cre (bottom) specimen. Scale bars=50 μm. (**B**) Differences in soma diameter of neurons based on isolation method. Grand mean ± SEM is plotted; per animal mean is denoted with colored circles (Int-Vgat=red, Int^IO-Vgat=maroon, Ntsr1=blue, and Int^RN-Ntsr1=navy). Int-Vgat (n=316 cells, 5 mice) or Int^IO-Vgat neurons (n=404 cells, 5 mice) are smaller than Int-Ntsr1(n=229 cells, 5 mice) or Int^RN-Ntsr1 neurons (n=125 cells, 4 mice; one-way ANOVA; Tukey's multiple comparison's test, p<0.0001****). (Note that Int includes all subdivisions of the interposed nucleus and icgs.) (**C**) Cumulative frequency distribution of measured cell diameter for all specimens. (**D**) The average proportion of the total (summed) RPS value that is derived from projections to motor, sensory, or modulatory extracerebellar brain regions. Mean and SEM are plotted. Welch's t-test with FDR correction of 1%, p=0.035 (ns), 0.0014 (**), 0.00023 (***),

*Figure 5 continued on next page*

Figure 5 continued

0.045 (ns), respectively. (**E**) Same as (**D**) but showing the contribution of ipsilateral or contralateral projections to total RPS per transgenic line. Welch's t-test with FDR correction of 1%, p=0.017 (ns) and 0.16 (ns), respectively. (**F**) Schematic of projection signatures from Ntsr1-Cre (blue) and Vgat-Cre (red). (**G**) Axons from Int-Vgat and Int-Ntsr1 follow unique paths through the pontine reticular nuclei (PRN). (**H**) Morphology differences in terminal contacts within the cerebellar cortex (top; boutons observed within the granule cell (GrC) layer; dotted white line in Nstr1 image denotes Purkinje Cell layer) and red nucleus (RN; bottom). Note mossy fiber nucleocortical terminals seen in Ntrs1-Cre mice but not Vgat-Cre mice. Scale bars=50 μm. Centromedial nucleus of the thalamus (CM), copula (Cop), Crus1 (Cr1) cuneate nucleus (CU), external cuneate nucleus (ECU), flocculus (Fl), gigantocellular reticular nucleus (GRN), hypoglossal nucleus (XII), inferior olive (IO), intermediate reticular nucleus (IRN), lateral reticular nucleus (LRN), lateral vestibular nucleus (LAV), medullary reticular nucleus, dorsal/ventral subdivision (MDRNd/v), midbrain reticular nucleus (MRN), Nucleus Y (Y), nucleus prepositus (PRP), oculomotor nucleus (III), parabrachial (PB), paracentral nucleus of the thalamus (PCN) parafasicular nucleus of the thalamus (PF), paraflocculus (PFl), paragigantocellular reticular nucleus, dorsal (PGRNd), paramedian lobule (PM), parvicellular reticular nucleus (PARN), periaqueductal grey (PAG), pontine grey (PG), pontine reticular nucleus (PRN), principal sensory nucleus of the trigeminal (PSV), simplex lobule (Sim), spinal trigeminal nucleus, caudal/interpolar subdivision (SPVc/i), spinal vestibular nucleus (SPIV), superior colliculus (SC), superior vestibular nucleus (SUV), supratrigeminal nucleus (SUT), superior vestibular nucleus (SUV), tegmental reticular nucleus (TRN), trigeminal motor nucleus (V), ventral tegmental area (VTA), ventromedial nucleus (VM), vestibular nuclei (VEST), and zona inderta (ZI).

The online version of this article includes the following source data for figure 5:

**Source data 1.** Raw data for **Figure 5B,C** containing the size (in pixels and microns) of thresholded YFP labeled soma in Int per specimen, data is segregated by transgenic mouse line.

neurons (**Liu et al., 2017**). 72.9±9.6% of starter cells in Vglut2-Cre specimens were mapped to Int (**Figure 6C and D**); with the remaining starter cells located in the lateral (15%), medial (2%), and superior vestibular nucleus (5%). Similarly, 80.8±4.7 % starter cells in Vgat-Cre mice were in Int, with the remainder in superior vestibular nuclei (7%), lateral (5%), medial (5%), and parabrachial (1%) nuclei (**Figure 6D**, **Supplementary file 4**). Total numbers of starter cell estimates (**Doykos et al., 2020**), defined by presence of rabies and TVA (**Figure 6C**) averaged 156±131 in Vglut2-Cre and 307±132 neurons in Vgat-Cre. TVA expression was not observed in cortex of Vgat-Cre or Vglut2-Cre mice, minimizing concerns of tracing contaminated by projections to cortical neurons.

The CbN receive a massive projection from Purkinje cells (PCs). The location of retrogradely labeled PCs was similar between specimens (**Figure 6E**), regardless of genotype. PCs in ipsilateral Lobules 4/5, Crus 1, and Simplex were most densely labeled following rabies starting in both cell types. No contralateral PC label was observed in any specimen.

Extracerebellar input to Vglut2-Cre and Vgat-Cre cells was diverse and wide-ranging (**Figure 6— figure supplement 1**). Both cell types receive input from brain regions related to motor, sensory, or modulatory functions (**Figure 6F**), corroborating previous observations with traditional tracers (**Fu et al., 2011**; but see **Barmack, 2003**). For a complete list of brain regions that provide input to Vglut2-Cre and Vgat-Cre Int neurons, see **Supplementary file 4** and **Figure 6—figure supplement 1**. Vglut2-Cre cells received a majority of inputs from ipsilateral sources, but not by large margins, with 64 % of inputs originating in ipsilateral regions (95% CI=[39, 90]; 36 % contralateral, 95% CI=[10, 61]). For Vgat-Cre cells, 54 % (95% CI=[50, 60]) of non-PC label was from ipsilateral sources (46 % contralateral, 95% CI=[40, 50]). These differences were not significant (t(2.2)=1.6, p=0.2; unpaired Welch's t-test). No extracerebellar region accounted for more than 10 % of the total cells, suggesting widespread integration within Int. Of note, significantly more LRN neurons were retrogradely labeled following Vgat-Cre injections (5.9 % of non-PC rabies labeled cells (95% CI=[3, 9]; >300 cells/specimen)) than Vglut2-Cre (0.4 % (95% CI=[−1.2, 1.9] of total non-PC rabies label cells (<40 cells/specimen; t(3.1)=7.6, p=0.004, unpaired Welch's t-test)), suggesting a more extensive input to Int-Vgat neurons from LRN. Aside from this difference, extracerebellar projections to both cell types came from medial vestibular nuclei, TRN, and other reticular formation nuclei. We observed retrograde label in the contralateral medial cerebellar nucleus from both Vgat and Vglut2-Cre mice.

Many canonical sources of mossy fibers, such as ECU, PRN, TRN, PG, and LRN (**Parenti et al., 1996**), were identified as sources of nuclear input as well as recipients of a projection from at least one cell type within Int (**Figure 7A–B**; **Tsukahara et al., 1983**; **Murakami et al., 1981**). **Figure 7B** summarizes the inputs and outputs of both cell types ranked by average proportion of total rabies labeled cells within a given region (for retrograde tracing, excluding Purkinje cells) or average projection strength (for anterograde tracing), excluding the weakest and least consistent projections, but including those that may originate from interstitial cell groups. The only brain regions which received a projection but were not also retrogradely labeled were the thalamic nuclei. In converse, the only brain regions with

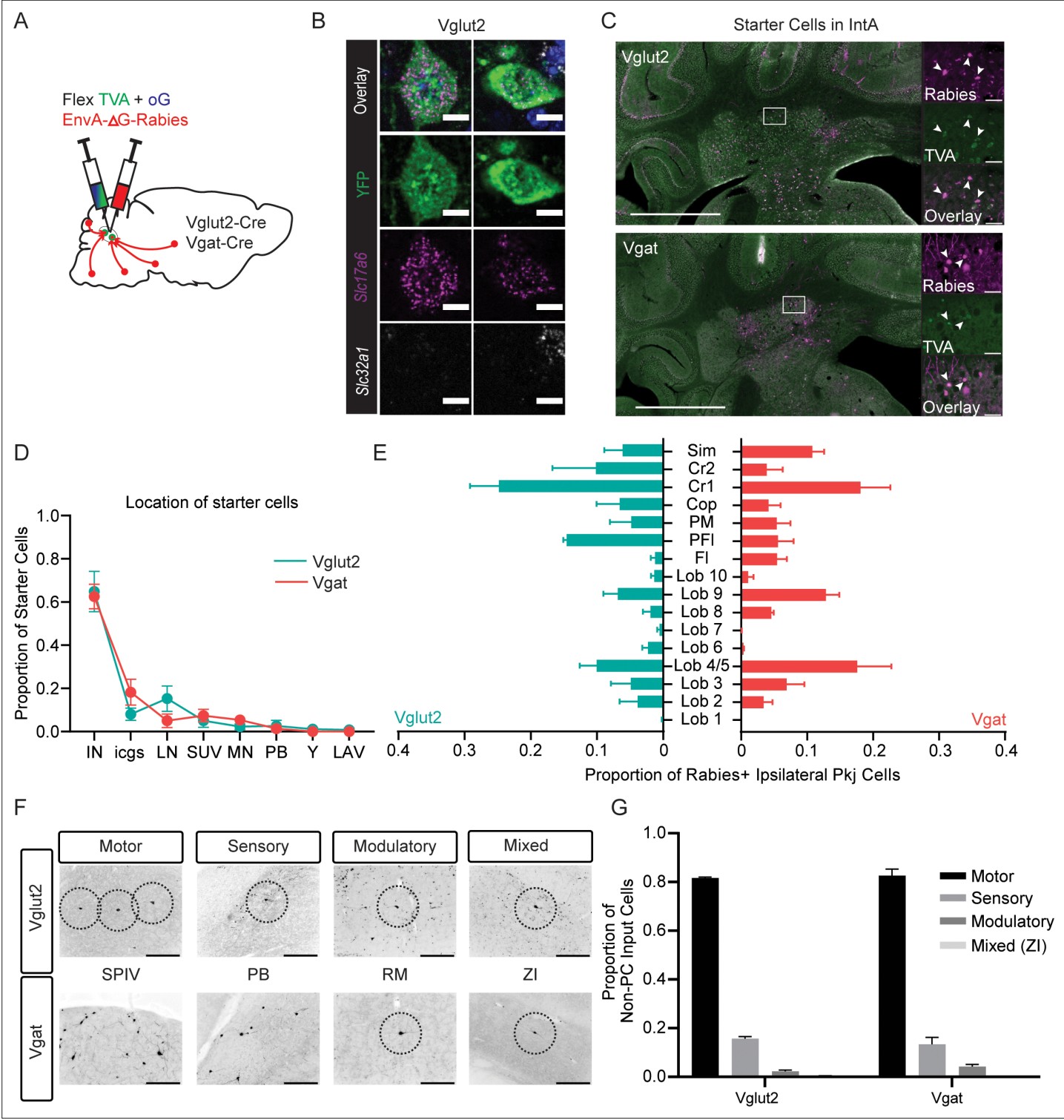

**Figure 6.** Monosynpatic tracing of inputs to the interposed nucleus. (**A**) Schematic of viral experiment. Cells labeled by this method provide monosynaptic input to Int. (**B**) Example *Slc17a6*-Cre driven YFP cell following in situ hybridization showing overlap with an mRNA probe against *Slc17a6* (Vglut2) and no overlap with an mRNA probe against *Slc32a1* (Vgat). Scale bars=10 μm. (**C**) Example starter cells from both transgenic mouse lines in IntA. Scale bars=1 mm. Insets to the right show rabies (magenta, top), TVA (green channel, top), and overlay (bottom). Scale bars=50 μm. (**D**) Locations of putative starter cells largely overlap for both cell types (mean + SEM). Note starter label in both IntA and IntP (IN). (**E**) Location of retrogradely labeled ipsilateral PCs by lobule. (**F**) Example extracerebellar rabies positive cells in motor (spinal vestibular nuclei, SPIV), sensory (parabrachial, PB), modulatory (raphe magnus, RM), and mixed (zona incerta, ZI) brain regions for both mouse lines. (**G**) Proportion of non-PC inputs to Vglut2-Cre or Vgat-Cre starter

*Figure 6 continued on next page*

*Figure 6 continued*

cells separated by modality. Simplex lobule (sim), Crus1 (Cr1), Crus 2 (Cr2), Copula (cop), paramedian lobule (PM), Paraflocculus (PFl), and Flocculus (FL).

The online version of this article includes the following figure supplement(s) for figure 6:

**Figure supplement 1.** Summary of monosynaptically labeled inputs to Vglut2 (teal, n=3 mice) and Vgat (red, n=3 mice) neurons in the interposed nucleus from extracerebellar regions.

retrogradely labeled cells but not anterograde projections were motor cortex, somatosensory cortex, subthalamic nucleus, and lateral hypothalamus, among other minor inputs (***Supplementary file 4***).

## Discussion

Here, we systematically examined the input and output patterns of diverse cell populations of the interposed cerebellar nucleus, Int, using intersectional viral tracing techniques. Consistent with previous work, we found that the putative excitatory output neurons of Int collateralize to regions of the contralateral brainstem, spinal cord, and thalamus and more restrictedly to the caudal ipsilateral brainstem, including to regions recently shown to control forelimb musculature. However, we also found that Int GABAergic projection neurons innervate brainstem regions other than IO, including the pontine nuclei, medullary reticular nuclei, and sensory brainstem structures. Interestingly, at least some IO-projecting neurons collateralize to comprise a subset of these projections. Inputs to these distinct cell types were also mapped using monosynaptic rabies tracing. We found that inputs to glutamatergic and Vgat neurons of the intermediate cerebellum are largely similar with only the LRN standing out as preferentially targeting Vgat neurons. Merging anterograde and retrograde datasets, region-level reciprocal loops between Int and brainstem targets were similar across both cell types.

The most surprising results were the diverse projections of GABAergic neurons of Int. To address concerns that these projections may be the result of a methodological artifact, we note a variety of data that support our interpretation. First, ISH and immunolabel support the view that Vgat-Cre is restricted to *Slc32a1* expressing neurons that express Gad65/67 in terminal boutons. Second, projection patterns of excitatory neurons were distinct, particularly within the ipsilateral caudal brainstem and diencephalon, thus non-specific viral label cannot account for the data. Third, AAV-retroCre injections into RN—a putative target of both Int-Vgat and Int-Ntsr—labeled targets matching mixed projections of excitatory and inhibitory neurons, including terminal label in IO. Finally, we used an intersectional approach, targeting Vgat-Cre expressing neurons that project to the IO. This method of isolating Int inhibitory neurons also consistently labeled terminals elsewhere in the brainstem. Taken together, leak of Cre cannot explain the sum of these observations.

Another study restricting tracer to lateral (dentate) nucleus Sox14-Cre expressing neurons, a transcription factor marking nucleo-olivary neurons, showed terminal label in the IO as well as the oculomotor nucleus (III). Based on retrograde tracing from III, terminals there were interpreted to reflect virus uptake by nucleus Y near the injection site (***Prekop et al., 2018***). This finding raised the question of whether brainstem and midbrain targets of Int-Vgat neurons described in the present study are merely a consequence of viral uptake in regions neighboring the interposed nucleus. Although projections were more extensive following larger injections in Vgat-Cre mice, we observed axon varicosities outside IO following injections that were completely restricted to the interposed nucleus. As has been noted in previous studies, the ventral border of the interposed nucleus is poorly distinguished but houses numerous islets of cells within the white matter tracts (***Sugihara and Shinoda, 2007***; ***Sugihara, 2011***). These regions receive Purkinje input from zebrin negative zones, and have been proposed to be distinct subregions of the CbN. A medial population, named the interstitial cell group, resides ventrally between the medial and interposed nuclei. An anterior extension, the anterior interstitial cell group, resides ventral to the interposed nucleus, and more posterior and laterally, the parvocellular interposed and lateral cell groups, neighboring nucleus Y, complete this constellation of loosely organized cell groups. Our data, which include injections of these regions, hint that these areas may house Vgat neurons that produce more extensive extra-IO projections, including nucleocortical beaded axons distinct from nucleocortical mossy fibers, although this conjecture will remain speculative until methodological advances permit cell-type-specific tracing from such minute regions to be carried out.

While these inhibitory projections were unknown, these data, combined with previous literature from the medial nucleus, suggest that inhibitory projections from the CbN may be a more prominent

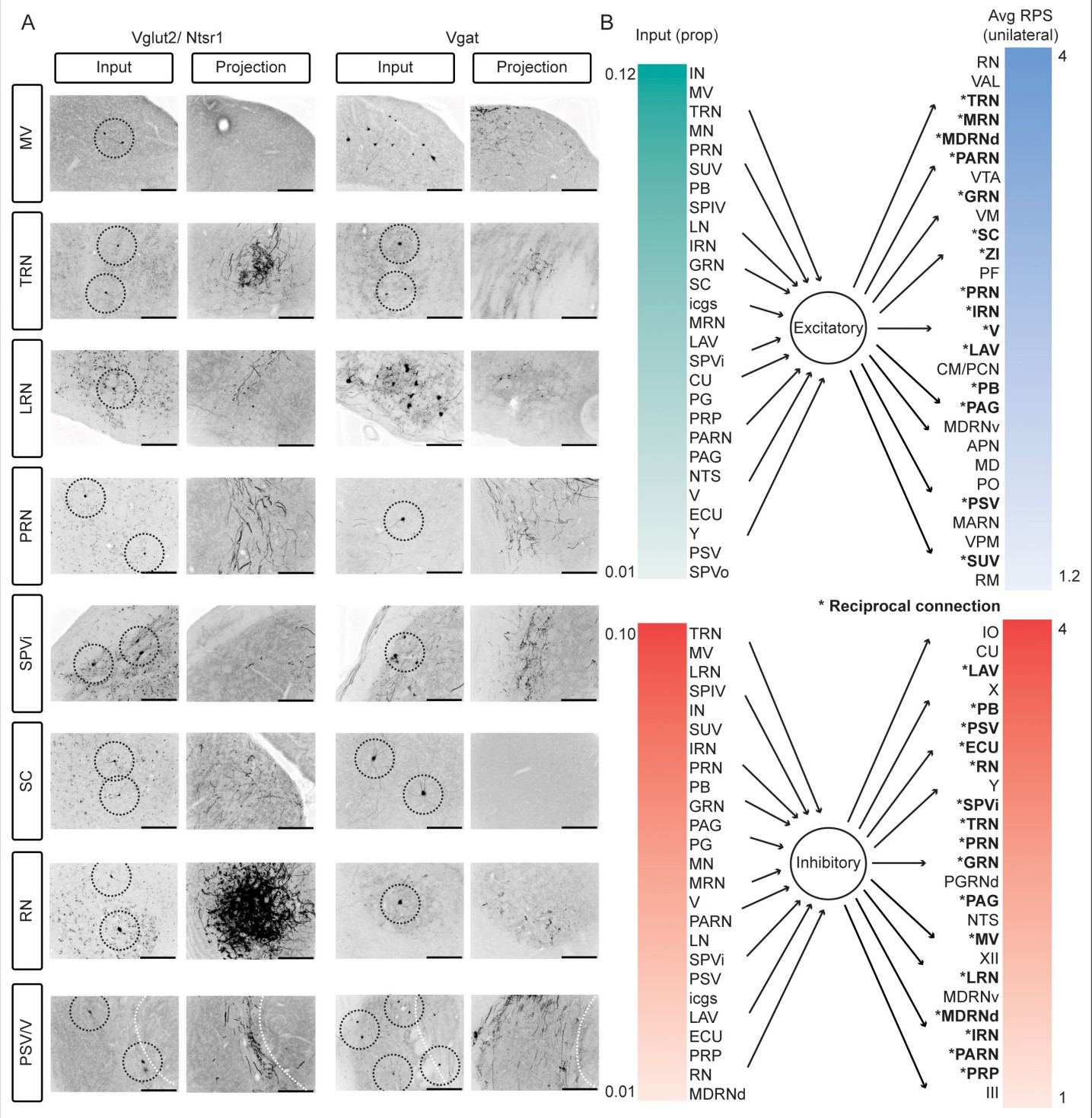

**Figure 7.** Reciprocal loops between Int and extracerebellar targets, for both Vglut2 and Vgat expressing cells. (**A**) Images depicting rabies labeled cells (columns 1 and 3, rabies + cells circled if singular or very small) and projections that included axon varicosities to the same regions at the same coordinates relative to bregma (columns 2 and 4). Medial vestibular nuclei (MV), tegmental reticular nucleus (TRN), lateral reticular nucleus (LRN), pontine reticular nuclei (PRN), spinal trigeminal nucleus, interpolar subdivision (SPVi), superior colliculus (SC), red nucleus (RN), principal sensory nucleus of the trigeminal (PSV), nd motor nucleus of the trigeminal (V). White dotted line denotes boundary between PSV and V. (**B**) Inputs and outputs listed in order of percent of non-PC rabies labeled cells (left) and relative projection strength (right). Only inputs with greater than 1 % of the total extracerebellar rabies labeled cells and regions with mean relative projection strengths greater than 1 are listed. Asterisks denote regions that constituted a major afferent (>1% of the total input) and received a major projection (an RPS >1 in *Slc32a1*-Cre mice and >1.2 in *Ntsr1*-Cre mice). Anterior pretectal nucleus

*Figure 7 continued on next page*

*Figure 7 continued*

(APN), centromedial nucleus of the thalamus (CM), cuneate nucleus (CU), external cuneate nucleus (ECU), gigantocellular reticular nucleus (GRN), hypoglossal nucleus (XII), inferior olive (IO), intermediate reticular nucleus (IRN), interposed nucleus (IN), interstitial cell groups (icgs), lateral nucleus (LN), lateral reticular nucleus (LRN), lateral vestibular nucleus (LAV), magnocellular reticular nucleus (MARN), medial nucleus (MN), medial vestibular nuclei (MV), mediodorsal nucleus of the thalamus (MD), medullary reticular nucleus, dorsal/ventral subdivision (MDRNd/v), midbrain reticular nucleus (MRN), motor nucleus of the trigeminal (V), nucleus of the solitary tract (NTS), nucleus raphe magnus (RM), nucleus X (X), Nucleus Y (Y), nucleus prepositus (PRP), oculomotor nucleus (III), parabrachial (PB), paracentral nucleus of the thalamus (PCN), parafasicular nucleus of the thalamus (PF), paragigantocellular reticular nucleus, dorsal (PGRNd), paramedian lobule (PM), parvicellular reticular nucleus (PARN), periaqueductal grey (PAG), pontine grey (PG), pontine reticular nucleus (PRN), principal sensory nucleus of the trigeminal (PSV), red nucleus (RN), spinal trigeminal nucleus, interpolar/oral subdivision (SPVi/o), spinal vestibular nucleus (SPIV), superior colliculus (SC), superior vestibular nucleus (SUV), supratrigeminal nucleus (SUT), superior vestibular nucleus (SUV), tegmental reticular nucleus (TRN), trigeminal motor nucleus (Y), ventral anterior-lateral complex of the thalamus (VAL), ventral posteromedial nucleus of the thalamus (VPM), ventral tegmental area (VTA), ventromedial nucleus (VM), and zona incerta (ZI).

circuit motif than is widely appreciated. The medial nucleus contains glycinergic projection neurons that innervate ipsilateral brainstem nuclei matching contralateral targets of excitatory neurons (*Bagnall et al., 2009*). Additional evidence of inhibitory outputs includes dual retrograde tracing suggesting that nucleo-olivary projections from medial nucleus and the vestibular complex collateralize to the VM hypothalamic nucleus (*Diagne et al., 2001*; *Li et al., 2017*). Studies combining retrograde horse radish peroxidase tracing from the basilar pontine nuclei (i.e., PG) with immunohistochemistry observed double-labeled GABA immunopositive neurons in the lateral nucleus of rats and cats (*Aas and Brodal, 1989*), although the literature is inconsistent (*Schwarz and Schmitz, 1997*). More recent work in mice tracing Vgat-Cre neurons of the lateral nucleus listed projections to a variety of brainstem structures as well as IO (*Locke et al., 2018*), but these results were not discussed.

Despite these corroborating experimental results, we note that our data may appear to contradict conclusions drawn from a dual-retrograde tracing study, in which only minor dual retrograde label was observed in the lateral and interposed nuclei following tracer injections into IO and RN or IO and TRN (*Ruigrok and Teune, 2014*). This study concluded that two distinct populations exist within the CbN: one which projects widely to several regions and one which projects exclusively to IO. However, this study did report a small number of cells colabeled by retrograde injections to IO and TRN as well as IO and RN. This observation may account for the present finding that a population of neurons that projects to both IO and premotor nuclei exists in smaller numbers, and that topographic specificity may have precluded previous methods from fully detecting the collateralization of inhibitory populations. Importantly, our results focus on all Int-Vgat neurons, and thus may label subsets of neurons that project exclusively to IO or exclusively outside of IO, which is not resolvable with dual retrograde methods.

Projection patterns of glycinergic medial and vestibular nucleus neurons have an ipsilateral bias relative to excitatory contralateral projections. (*Bagnall et al., 2009*; *Prekop et al., 2018*; *Sekirnjak et al., 2003*; *Shin et al., 2011*). This organizational structure has been proposed to potentially mediate axial muscular opponency. While there was also a trend for an ipsilateral targeting of Int-Vgat neurons, this bias was not significant when accounting for false discovery rates (p=0.02), with both excitatory and inhibitory cells projecting bilaterally. Future studies investigating the functional roles of these projections may explore agonist/antagonist opponency in motor targets of these projections, which remain lateralized for limb musculature. Additionally, the widespread observation of Purkinje neurons that increase rates during cerebellar dependent behaviors may suggest the potential for a double disinhibitory pathway through the CbN, if these Purkinje neurons converged on inhibitory nuclear output neurons (*De Zeeuw and Berrebi, 1995*; *Zeeuw and Chris, 2020*).

What might be the role of inhibitory projections from the CbN? Two intriguing patterns emerged that are suggestive of potential function. First, inhibitory projections targeted more sensory brainstem structures than excitatory outputs. Predicting sensory consequences of self-generated movement, termed forward models, is a leading hypothesis for the role of cerebellum in sensorimotor behaviors. While populations of Purkinje neurons may perform this computation, it is unknown how forward models are used by downstream targets. Inhibitory projections from cerebellum to sensory areas would seem to be ideally situated to modulate the sensory gain of predicted sensory consequences of movement (*Brooks et al., 2015*; *Shadmehr, 2020*). Moreover, negative sensory prediction error could be used to actively cancel predicted sensory reafference (*Kim et al., 2020*; *Requarth and Sawtell, 2014*; *Shadmehr, 2020*; *Conner et al., 2021*), raising implications for a combined role

of negative sensory prediction error in guiding learning both through modulation of climbing fiber signaling in IO and through modulation of sensory signals reaching the cerebellum upon which associative learning is built. Second, GABAergic projections to the pontine nuclei, which are themselves a major source of mossy fiber inputs to the cerebellum, suggest a regulatory feedback pathway that could operate as a homeostat akin to the feedback loops through the IO (*Medina et al., 2002*). The pontine nuclei are a major relay of cortical information into the cerebellum. Thus, through inhibitory feedback, cortical information could potentially be gated to facilitate strategic (i.e., cortical) control for novel skill learning or turned down to facilitate automatic (i.e., ascending/non-cortical) control of movements (*Schwarz and Thier, 1999*).

The present study compliments a recent collection of papers examining cerebellar nuclear cell types. Transcriptomics analyses of the CbN identified three distinct excitatory cell types within IntA. These classes included two broad projection types: those that target a wide array of brainstem nuclei and those that target the ZI (*Kebschull et al., 2020*). Another recent study identified two distinct interposed cell types based on projection patterns to the spinal cord, which were shown to constitute a minority of neurons (<20%). Nevertheless, these spinal-projecting neurons collateralized to many other targets, including the MDRNv, RN, and the VAL (*Sathyamurthy et al., 2020*). Inhibitory projections were not examined in these studies, thus it will be interesting to examine how the inhibitory projection neurons identified in the present study map onto transcript clusters of the inhibitory cell types, five total across the CbN. At a minimum, these clusters would include IO-projecting neurons, interneurons, MN glycinergic projection neurons, and a collateralizing population of inhibitory neurons identified here (*Ankri et al., 2015*; *Bagnall et al., 2009*; *Fujita et al., 2020*; *Husson et al., 2014*; *Kebschull et al., 2020*; *Sathyamurthy et al., 2020*).

Inputs to these neuronal populations were largely similar, though we observed minor differences in the input signatures of Int-Vglut2 and Int-Vgat. Many more neurons in the LRN were labeled following Vgat-Cre starting cells for monosynaptic rabies tracing, suggesting a predominant innervation of inhibitory neurons by LRN. It remains unclear if there are differences in input connectivity between Vgat subgroups, specifically interneurons and projection neurons, or whether Gad65/67 expressing neurons co-express GlyT2. In comparing input and outputs to diverse cell types, we noticed that reciprocal loops were common, broadening themes of reciprocal loops demonstrated previously (*Tsukahara et al., 1983*; *Beitzel et al., 2017*; *Murakami et al., 1981*), to also include inhibitory neurons. Such loops resemble neural integrators used in gaze maintenance or postural limb stabilization (*Albert et al., 2020*; *Cannon and Robinson, 1987*), another potential functional role of the anatomy presented here. Interestingly, we observed a few neocortical inputs to the intermediate/interstitial groups of the CbN. We speculate that these regions may conform to the reciprocal loop motif, albeit polysynaptically, predicting that thalamic targets innervate neocortical areas that project back to the CbN.

In conclusion, the anatomical observations presented here open the door to many potential functional studies that could explore the roles of inhibitory projections in real-time motor control, sensory prediction and cancellation, and dynamic cerebellar gain control. Taken together, the present results suggest distinct computational modules within the interposed CbN based on cell types and shared, but likely distinct, participation in motor execution.

## Materials and methods

### Key resources table

| Reagent type (species) or resource | Designation | Source or reference | Identifiers | Additional information |
|---|---|---|---|---|
| Strain, strain background (*Mus musculus*) | C57BL/6 J | Charles River | Stock | |
| Genetic reagent (*M. musculus*) | Gad1-Cre | Gift from Dr. Diego Restrepo, recv'd frozen embryos from Tamamaki group | RRID: IMSR CARD:2065 | PMID:19915725 |

*Continued on next page*

*Continued*

| Reagent type (species) or resource | Designation | Source or reference | Identifiers | Additional information |
|---|---|---|---|---|
| Genetic reagent (*M. musculus*) | Ntsr1-Cre | Mutant Mouse Regional Resource Center | RRID: MMRRC_030648-UCD Stock, Tg(Ntsr1-cre) GN220 Gsat/Mmucd | PMID:17855595 |
| Genetic reagent (*M. musculus*) | Vgat-ires-cre knock-in (C57BL/6 J) | Jackson Labs | RRID: IMSR_JAX:028862 Stock, #028862 | PMID:21745644 |
| Genetic reagent (*M. musculus*) | Vglut2-ires-cre knock-in (C57BL/6 J) | Jackson Labs | RRID: IMSR_JAX:028862 Stock, #028863 | PMID:21745644 |
| Recombinant DNA reagent | AAV1.CAG.flex.GFP (virus) | Addgene | RRID: Addgene_51502 Lot #: V41177 | Titer: $2.0 \times 10^{13}$ (GFP) $1.2 \times 10^{13}$ (RFP) |
| Recombinant DNA reagent | AAV1.CAG.flex.RFP (virus) | Addgene | RRID: Addgene_28306 Lot #: V5282 | |
| Recombinant DNA reagent | rAAV2.EF1a.DIO. eYFP.WPRE.pA (virus) | UNC | RRID: Addgene_27056 Lot #: AV4842F | Titer: $4.5 \times 10^{12}$ |
| Recombinant DNA reagent | AAV8.hysn.ConFon. eYFP (virus) | Addgene | RRID: Addgene_55650 Lot #: V15284 | PMID:24908100 Titer: $2.97 \times 10^{13}$ |
| Recombinant DNA reagent | AAVretro.EF1a. FlpO (virus) | Addgene | RRID: Addgene_55637 Lot # V56725 | PMID:24908100 Titer: $1.02 \times 10^{13}$ |
| Recombinant DNA reagent | AAV2.retro.hSyn. NLS.GFP.Cre (virus) | Viral preparations were a gift of Dr. Jason Aoto | RRID: Addgene_175381 | PMID:23827676 |
| Recombinant DNA reagent | AAV9.EF1a.FLEX. H2B.GFP.2A.oG (virus) | Salk Institute | RRID: Addgene_74289 | Titer: $2.41 \times 10^{12}$ |
| Recombinant DNA reagent | AAV1.EF1a.FLEX. TVA.mCherry (virus) | UNC | RRID: Addgene_38044 | PMID:22681690 |
| Recombinant DNA reagent | AAV1.syn.FLEX.split TVA.EGFP.tTA (virus) | Addgene | RRID: Addgene_100798 | PMID:28847002 |
| Recombinant DNA reagent | AAV1.TREtight.mTag BFP2.B19G (virus) | Addgene | RRID: Addgene_100799 | PMID:28847002 |
| Recombinant DNA reagent | EnvA.Gdeleted. EGFP (virus) | Salk Institute | RRID: Addgene_32635 | |
| Recombinant DNA reagent | EnvA.Gdeleted. mCherry (virus) | Salk Institute | RRID: Addgene_32636 | |
| Antibody | Anti-Vglut2 (Rabbit monoclonal) | Abcam | RRID: AB_2893024 Cat #: a FP1487001KT b216463 | Lot #: GR3249111-2 (1:250) |
| Antibody | Anti-Gad65/67 (Rabbit polyclonal) | Sigma-Aldrich | RRID: AB_2893025 Cat#: ABN904 | Lot#: 3384833 (1:200) |
| Antibody | Anti-Tyrosine Hydroxylase (Sheep polyclonal) | MilliporeSigma | RRID: AB_90755 Cat#: AB1542 | (1:200) |
| Antibody | Anti-GFP-Alexa Fluor 488 conjugate (Rabbit polyclonal) | Invitrogen | RRID: AB_221477 Cat#: A21311 | Lot #: 2017366 (1:400) |
| Antibody | Anti-Rabbit DyL594 (Goat polyclonal) | Bethyl | RRID:AB_10631380 Cat#: A120-601D4 | (1:400) |
| Antibody | Anti-Mouse AF555 (Goat polyclonal) | Life Technologies | RRID:AB_141596 Cat#: A21127 | (1:400) |
| Antibody | Anti-Sheep AF 568 (Donkey polyclonal) | Life Technologies | RRID:AB_2535753 Cat#: A21099 | (1:400) |

*Continued*

| Reagent type (species) or resource | Designation | Source or reference | Identifiers | Additional information |
|---|---|---|---|---|
| Commercial Assay Kit | RNAscope Intro Pack for Multiplex Fluorescent Reagent Kit v2 | Advanced Cell Diagnostics | Cat#: 323,136 | |
| Sequence-based reagent | EYFP-C1 | Advanced Cell Diagnostics | Cat#: 312,131 | mRNA probe |
| Sequence-based reagent | Mm-Slc32a1-C2 | Advanced Cell Diagnostics | Cat#: 319,191 | mRNA probe |
| Sequence-based reagent | Mm-Slc17a6-C3 | Advanced Cell Diagnostics | Cat#: 319,171 | mRNA probe |
| Other | Opal Fluorophore Reagent Pack 520 | Akoya Biosciences | Cat #: FP1487001KT | |
| Other | Opal Fluorophore Reagent Pack 570 | Akoya Biosciences | Cat #: FP1488001KT | |
| Other | Opal Fluorophore Reagent Pack 690 | Akoya Biosciences | Cat #: FP1497001KT | |
| Other | DAPI stain | Advanced Cell Diagnostics | Cat#: 323,108 | |

## Animals

All procedures followed the National Institutes of Health Guidelines and were approved by the Institutional Animal Care and Use Committee at the University of Colorado Anschutz Medical Campus. Animals were housed in an environmentally controlled room, kept on a 12 hr light/dark cycle, and had ad libitum access to food and water. Adult mice of either sex were used in all experiments. Genotypes used were C57BL/6 (Charles River Laboratories), Neurotensin receptor1-Cre [Ntsr1-Cre; Mutant Mouse Regional Resource Center, STOCK Tg(Ntsr1-Cre) GN220Gsat/Mmucd], Gad1-Cre (*Higo et al., 2009*); *Slc32a1* (Vgat)-Cre[#028862; Jackson Labs], and *Slc17a6* (Vglut2)-Cre [#028863; Jackson Labs]. All transgenic animals were bred on a C57BL/6 background. Gad1 and Ntsr1-Cre mice were maintained as heterozygotes and were genotyped for Cre (Transnetyx). For all surgical procedures, mice were anesthetized with intraperitoneal injections of a ketamine hydrochloride (100 mg/kg) and xylazine (10 mg/kg) cocktail, placed in a stereotaxic apparatus, and prepared for surgery with a scalp incision.

## Viral injections

Injections were administered using a pulled glass pipette. Unilateral pressure injections of 70–200 nl of Cre-dependent reporter viruses (AAV1-CAG-flex-GFP; AAV2-DIO-EF1a-eYFP; AAV8-hysn-ConFon-eYFP, see Key Resources Table) were made into Int. Injections were centered on IntA, with minor but unavoidable somatic label appearing in posterior interposed (IntP), lateral nucleus (LN), interstitial cell groups (icgs), and the dorsal region of the vestibular (VEST) nuclei, including dorsal portions of the superior vestibular nucleus (SUV), lateral vestibular nucleus (LAV), and Nucleus Y (Y). We occasionally observed minor somatic label in the parabrachial nucleus (PB) and the cerebellar cortex (Cb-Ctx) anterior or dorsal, respectively, to Int in injections into Vgat-Cre mice. In control injections (n=3; virus in C57/Bl6 mice or off-target injection into Ntsr-1 Cre mice), viral expression was not detected. We did not see appreciable somatic label in the medial nucleus (MN) of any specimens. For RN injections, craniotomies were made unilaterally above RN (from bregma: 3.5 mm posterior, 0.5 mm lateral, and 3.6 mm ventral). For Int injections, unilateral injections were made at lambda: 1.9 mm posterior, 1.6 mm lateral, and 2.1 mm ventral. For IO injections, the mouse's head was clamped facing downward, an incision was made near the occipital ridge, muscle and other tissue was removed just under the occipital ridge, and unilateral injections were made at 0.2 mm lateral, and 2.1 mm ventral with the pipet tilted 10° from the Obex. This method consistently labeled IO and had the advantage of avoiding accidental cerebellar label via pipette leakage. To achieve restricted injection sites, smaller volumes were required in Vgat-Cre mice compared to Ntsr1-Cre mice (40–100 nl vs. 150–200 nl, respectively). The smallest Vgat-Cre injection was made iontophoretically using 2 M NaCl. Current (5 µA) was applied for 10 min, the current was removed and after a waiting period of 5 min, the pipet was retracted. Retrograde labeling of RN-projecting IntA neurons was achieved through AAVretro-EF1a-cre (*Tervo et al., 2016*) Retrograde injections of RN were performed simultaneously with flex-GFP injections of IntA.

Retrograde virus (AAVretro-EF1a-Flp) was injected into IO 1 week before reporter viruses because of the different targeting scheme and mice were allowed to heal 1 week prior to the reporter virus injection. All mice injected with AAVs were housed postoperatively for ~6 weeks before perfusion to allow for viral expression throughout the entirety of the axonal arbor. Control injections were performed where Cre or Flp expression was omitted, either by performing the injections in wild-type mice or in transgenic mice without the Retro-flp injection into IO or RN, confirming the necessity of recombinase presence in reporter expression (*Fenno et al., 2017*).

For monosynaptic rabies retrograde tracing, AAV1-syn-FLEX-splitTVA-EGFP-tTA and AAV1-TREtight-mTagBFP2-B19G (Addgene; *Liu et al., 2017*) were diluted 1:200 and 1:20, respectively, and mixed 1:1 before co-injecting (100 nl of each; vortexed together) unilaterally into IN of *Slc32a1*-IRES-Cre (n=3; "Vgat") and *Slc17a6*-IRES-Cre (n=1; "Vglut2") mice. Two additional Vglut2-Cre mice were prepared using AAV1-EF1a-Flex-TVA-mCherry (University of North Carolina Vector Core; *Watabe-Uchida et al., 2012*) and AAV9-Flex-H2B-GFP-2A-oG (Salk Gene Transfer, Targeting and Therapeutics Core; *Kim et al., 2016*). After a 4–6 week incubation period, a second injection of EnvA-SADΔG-eGFP virus (150–200 nl) was made at the same location (Salk Gene Transfer, Targeting and Therapeutics Core; *Kim et al., 2016*; *Wall et al., 2010*; *Wickersham et al., 2007*). Mice were sacrificed 1 week following the rabies injection and prepared for histological examination. Control mice (C57Bl/6; n=1) were injected in the same manner, however, without Cre, very little putative Rabies expression was driven, though eight cells were noted near the injection site. No cells were identified outside this region (*Figure 2—figure supplement 1*).

## Tissue preparation and imaging

Mice were overdosed with an intraperitoneal injection of a sodium pentobarbital solution, Fatal Plus (MWI), and perfused transcardially with 0.9 % saline followed by 4 % paraformaldehyde in 0.1 M phosphate buffer. Brains were removed and postfixed for at least 24 hr then cryoprotected in 30 % sucrose for at least 24 hr. Tissue was sliced in 40 μm consecutive coronal sections using a freezing microtome and stored in 0.1 M phosphate buffer. Sections used for immunohistochemistry were floated in phosphate-buffered saline (PBS), permeabilized using 0.1–0.3% Triton X-100, placed in blocking solution (2–10% Normal Goat serum depending on antibody) for 1–2 hr, washed in PBS, and incubated in primary antibodies GFP (1:400), Gad65/67 (1:200), Vglut2(1:250), and TH (1:200) for 24–72 hr. Sections were then washed in PBS three times for a total of 30 min before incubation in secondary antibodies (Goat anti-rabbit DyL594, Goat anti-mouse AF555 (1:400), or Goat anti-sheep AF568 (1:400), see Key Resources Table) for 60–90 min. Finally, immunostained tissue was washed in PBS and mounted in Fluoromount G (SouthernBiotech).

Every section for rabies experiments and every third section for anterograde tracing experiments was mounted onto slides and imaged. Spinal cord sections were also sliced in 40 μm consecutive coronal sections with every fourth section mounted. Slides were imaged at 10 × using a Keyence BZX-800 epifluorescence microscope or a slide-scanning microscope (Leica DM6000B Epifluorescence & Brightfield Slide Scanner; Leica HC PL APO 10 × Objective with a 0.4 numerical aperture; Objective Imaging Surveyor, V7.0.0.9 MT). Images were converted to TIFFs (OIViewer Application V9.0.2.0) and analyzed or adjusted via pixel intensity histograms in ImageJ. We inverted fluorescence images using greyscale lookup tables in order to illustrate results more clearly. YFP terminals stained for neurotransmitter transport proteins (Gad65/67 or Vglut2) were imaged using a 100 × oil objective on a Marianas Inverted Spinning Disc confocal microscope (3I). We imaged in a single focal plane of 0.2 μm depth to analyze colocalization of single terminal endings. Images were analyzed in ImageJ.

## Analysis of overlap by genetically defined neurons

To distinguish overlap of Cre expression with transmitter markers, we performed ISH. For ISHs, RNAse-free PBS was used for perfusions and the tissue was cryoprotected by serial applications of 10%, 20%, and 30 % Sucrose for 24–48 hr each. The brain tissue was then embedded in OCT medium and sliced to 14 μm thick sections on a cryostat (Leica HM 505 E). Tissue sections were collected directly onto SuperFrostPlus slides and stored at –80 °C for up to 3 months until RNA ISH for EYFP (virally driven), *Slc32a1* (Vgat), and *Slc17a6* (Vglut2) from RNAscope Multiplex Fluorescent Reagent Kit v2 (Advanced Cell Diagnostics). The slides were defrosted, washed in PBS, and baked for 45 min at 60 °C in the HybEZ oven (Advanced Cell Diagnostics) prior to post-fixation in 4 % PFA for 15 min at 4 °C and

dehydration in ethanol. The sections were then incubated at room temperature in hydrogen peroxide for 10 min before performing target retrieval in boiling 1 × target retrieval buffer (Advanced Cell Diagnostics) for 5 min. The slides were dried overnight before pretreating in protease III (Advanced Cell Diagnostics) at 40 °C for 30 min. The RNAscope probes #312131, #319191, and #319171 were applied and incubated at 40 °C for 2 hr. Sections were then treated with preamplifier and amplifier probes by applying AMP1, AMP2 at 40 °C for 30 min and AMP3 at 40 °C for 15 min. The HRP signals were developed using Opal dyes (Akoya Biosciences): 520 (EYFP probe), 570 (*Slc17a6* probe), and 690 (*Slc32a1* probe) and blocked with HRP blocker for 30 min each.

The CbN were stained using DAPI for 30 s before mounting in Prolong Gold (Thermo Fisher Scientific). Washes were performed two times between steps using 1 × wash buffer (Advanced Cell Diagnostics). Fluorescence was imaged for YFP, *Slc17a6* (Vglut2), *Slc32a1* (Vgat), and DAPI using a Zeiss LSM780 microscope. Each image was captured using a 34-Channel GaAsP QUASAR Detection Unit (Zeiss) at 40 × magnification in water from 14 µm sections. Images were stitched using ZEN2011 software and analyzed in ImageJ. Cre-expressing cells were identified by somatic labeling in the YFP channel; colocalization with the *Slc32a1* (Vgat) or *Slc17a6* (Vglut2) channel was determined by eye using a single composite image and the 'channels tool.' Positive (Advanced Cell Diagnostics, PN 320881) and negative control probes (Advanced Cell Diagnostics, PN 32087) resulted in the expected fluorescent patterns (*Figure 1—figure supplement 3*).

We analyzed the fidelity of our transgenic lines using virally mediated YFP somatic label and DAPI staining to identify cells expressing Cre and analyzed the colocalization of *Slc32a1* or *Slc17a6* mRNA within the bounds of a YFP cell. The YFP signal was often less punctate than other endogenous mRNA probes, thus we restricted our analysis to cells largely filled by YFP signal that contained DAPI stained nuclei. Due to the high expression patterns of *Slc32a1* and *Slc17a6*, analyzing by eye was reasonable. Only a total of four cells across all analyzed sections appeared to have ISH-dots in both the *Slc32a1* and *Slc17a6* channels. This may be due to poor focus on these individual cells or background fluorescence. Two sections per animal and one (Vglut2-Cre and Gad1-Cre) or two animals (Ntsr1-Cre and Vgat-Cre) per transgenic line were counted.

In preliminary studies, we tested a Gad1-Cre driver line (*Higo et al., 2009*) for specificity since *Gad1* was recently identified as a marker of inhibitory neurons within the CbN (*Kebschull et al., 2020*). However, in this line, we observed clear instances of both Gad65/67 and Vglut2- immunoreactivity in YFP labeled terminal varicosities as well as some *Slc17a6* mRNA (Vglut2) expression in YFP expressing somata (*Supplementary file 2*, *Figure 1—figure supplements 3–4*). 87 % YFP expressing cells colocalized with *Slc32a1* (60/69 cells) and 13 % colocalized with *Slc17a6* (9/60 cells). The clear instances of promiscuity in Gad1-Cre mice precluded further use of these mice in the present study.

## Cell size analysis

We imaged cells within IntA at 20 × with the epifluorescent Keyence microscope, then used the 'Measure' tool in ImageJ to gather the cross-sectional area and the 'Fit ellipse' measurement to gather minimum and maximum diameter which we converted from pixels to microns using reference scale bars. We report the maximum diameter. We analyzed 15–110 well focused and isolated cells for each specimen. Statistical analyses were conducted using a repeated-measures ANOVA on the per-animal and grand means of cell diameter per experimental condition.

## Brain region classification

We used a combination of the Allen Mouse Brain Reference Atlas and the *Mouse Brain in Stereotaxic Coordinates* by Franklin and Paxinos to identify brain regions, while noting that there were minor differences in location, shape, and naming of the brain regions between these reference sources (*Lein et al., 2007*; *Paxinos, 2008*). We use the term IntA to refer to the anterior and dorsolateral interposed nucleus; IntP to refer to the posterior interposed nucleus; IN in the transsynaptic tracing section to refer to all subdivisions of the interposed nucleus; and Int to refer to the intermediate region of the deep CbN including all subdivisions of the interposed nucleus and the more ventral intermediate interstitial cell groups. We grouped the dorsolateral and anterior subdivisions of Int because they were often co-labeled, are difficult to confidently distinguish, and occur at similar anterior-posterior (AP) coordinates. For monosynaptic rabies tracing and difficulty in targeting multiple viruses to the exact same location—we grouped all subdivisions of the interposed nucleus (IN; anterior, posterior,

and dorsolateral). In general, we followed nomenclature and coordinates respective to bregma of the Allen Mouse Brain Reference Atlas including its classification conventions of motor, sensory, modulatory/affective sources from the 2008 version. Thalamic regions were classified as motor if they project to motor cortices; sensory if they project to sensory cortices, with intralaminar thalamic nuclei classified as modulatory/affective. The intermediate and deep layers of the SC harbored terminal fields and retrogradely labeled neurons and is thus classified as motor.

## Projection quantification

All specimens were included in data sets if their injection sites were centered on IntA, with spillover described in each figure. If no label was observed or injections were mistargeted, specimen were excluded from further consideration. We mapped terminals to a collection of extracerebellar targets spanning the anterior-posterior (A-P) axis from the posterior medulla to the thalamus. We assigned terminal fields a semi-quantitative relative projection strength (RPS) of 0–4 based on the density and anterior-posterior spread (*Supplementary file 1*) made by a single observer and verified by a second. The values were assigned relative to the highest density projection target for each genotype: All Ntsr1-Cre projection fields were assessed relative to the density of terminals in RN whereas Vgat-Cre specimens were assessed relative to the density of IO terminals (*Figure 1—figure supplement 1*). Briefly, a terminal field that was both dense and broad (in spanning the anterior-posterior axis) was assigned a RPS of 4, semi-dense and semi-broad assigned a 3, semi-dense and/or semi-broad a 2, and fields determined to be sparse but nevertheless present, were assigned an RPS of 1. In addition, we compared our specimens to analogous preparations published in the Allen Mouse Brain Connectivity Atlas, specifically the histological profile of Cre-dependent labeling following injections into IntA of either Ntsr1-Cre or Vgat-Cre mice. These publicly available sources recapitulated projection signatures from lab specimens (*Supplementary file 2*). We included the Allen injection data in our analysis of average projection strength for Ntsr1-Cre (n=1) and Vgat-Cre (n=1) specimen but did not use the histological images of these injections here. The full histological profiles of genetically restricted GFP label from the Allen can be found at: 2011 Allen Institute for Brain Science. Mouse Brain Connectivity Atlas. Available from: http://connectivity.brain-map.org/, experiments #264096952, #304537794.

We determined the average proportion of the total RPS value that is derived from specific projections (to specific modalities or hemispheres; *Figure 5D and E*) by summing the RPS values to every region receiving a projection per specimen. We then divided this number by summated RPS values in the groupings of interest. We report the average proportion of total RPS values across all specimens in each experimental cohort (biological replicates). These measurements are therefore indicative of the strength of projection to certain modalities or hemispheres, and not simply a measure of the number of brain regions targeted.

## Rabies quantification

We identified presumptive starter cells as rabies (mCherry) positive cells that also contained GFP (AAV1-syn-FLEX-splitTVA-EGFP-tTA). We used an antibody against EGFP (see Key Resources Table) to visualize TVA at these concentrations, but mBFP (AAV1-TREtight-mTagBFP2-B19G) could not be visualized. However, oG expression is restricted to cells expressing TVA due to the necessity of the tetracycline transactivator gene encoded by the virus delivering TVA (*Liu et al., 2017*). In two Vglut2-Cre mice used for rabies tracing, we identified presumptive starter cells as rabies positive cells within the CbN where both mCherry (AAV1-EF1a-Flex-TVA-mCherry) and GFP (AAV9-Flex-H2B-GFP-2A-oG-GFP/ EnvA-SADΔG-eGFP) were expressed. We could not easily identify cells in which all three components were present due to overlapping fluorescence from the oG and modified rabies viruses, thus starter cell identification is an estimate (*Doykos et al., 2020*).

## Acknowledgements

We thank Courtney Dobrott for sharing expertise in spinal cord removal, RNAscope methodology, and helpful comments on the manuscript. We thank Aya Miften for assistance in histology and Daniel Heck for work on preliminary datasets. We are grateful to Dr Jason Aoto for the preparation of AAVRetro viruses. This work was supported by NS114430, NSF CAREER 1749568, and by a grant from the Simons Foundation as part of the Simons-Emory International Consortium on Motor Control. Imaging of rabies label was performed in the Advanced Light Microscopy Core, part of the NeuroTechnology

Center at University of Colorado Anschutz Medical Campus supported in part by Rocky Mountain Neurological Disorders Core Grant Number P30 NS048154 and by Diabetes Research Center Grant Number P30 DK116073 with the assistance of Dr. Radu Moldovan. Rabies viruses were obtained from the Salk GT3 Core Facility supported by NIH-NCI CCSG: P30 014195, an NINDS R24 Core Grant and funding from NEI.

# Additional information

### Funding

| Funder | Grant reference number | Author |
|---|---|---|
| National Institute of Neurological Disorders and Stroke | 114430 | Abigail L Person |
| National Science Foundation | 1749568 | Abigail L Person |
| Simons Foundation | | Abigail L Person |

The funders had no role in study design, data collection and interpretation, or the decision to submit the work for publication.

### Author contributions

Elena N Judd, Conceptualization, Data curation, Formal analysis, Investigation, Visualization, Writing - original draft; Samantha M Lewis, Data curation, Investigation, Methodology; Abigail L Person, Conceptualization, Funding acquisition, Investigation, Methodology, Project administration, Supervision, Validation, Writing - original draft, Writing - review and editing

### Author ORCIDs

Elena N Judd (iD) http://orcid.org/0000-0001-8211-5140
Abigail L Person (iD) http://orcid.org/0000-0001-9805-7600

### Ethics

All procedures followed the National Institutes of Health Guidelines and were approved by the Institutional Animal Care and Use Committee at the University of Colorado Anschutz Medical Campus under protocol #43, Laboratory of Abigail Person, re-approved 11/2020. Every effort was made to minimize suffering.

### Decision letter and Author response

Decision letter https://doi.org/10.7554/eLife.66231.sa1
Author response https://doi.org/10.7554/eLife.66231.sa2

# Additional files

### Supplementary files

• Transparent reporting form
• Supplementary file 1. Summary of average projection strength by targeting method.
• Supplementary file 2. In situ hybridization results.
• Supplementary file 3. Anterograde tracing data by specimen.
• Supplementary file 4. Rabies positive neurons in each brain region by specimen.
• Source data 1. Projection strength data by targeting method.
• Source data 2. Analysis of in situ hybridization data.
• Source data 3. Proportion of rabies labeled cells in Vgat and Vglut2-cre mice.

### Data availability

All data analysis is included in the manuscript and supporting files.

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
