## [Decision Letter]

**Acceptance summary:**

There is a growing body of evidence demonstrating that the cerebellum is involved in motor and non-motor functions. In this paper, Judd and colleagues present compelling evidence that cerebellar output neurons are more diverse in their projections than previously appreciated. They show that excitatory and inhibitory cerebellar nuclei neurons innervate a range of extra-cerebellar loci. The anatomical maps uncovered in this work could support diverse cerebellar functions that control different behaviors.

**Decision letter after peer review:**

Thank you for submitting your article "Widespread inhibitory projections from the interposed cerebellar nucleus" for consideration by *eLife*. Your article has been reviewed by 3 peer reviewers, and the evaluation has been overseen by Roy V Sillitoe as Reviewing Editor and Reviewer #1, and Ronald Calabrese as the Senior Editor. The following individuals involved in review of your submission have agreed to reveal their identity: Albert I Chen (Reviewer #3); Hirofumi Fujita (Reviewer #4).

Summary:

In this study, Judd and colleagues use a combination of mouse genetics and viral marking to expand the extra-cerebellar map of projections. These data will impact our understanding of how the cerebellum contributes to behavior and in general how brain function is packaged at the anatomical level. These data will not only impact cerebellar scientists but also those workers interested in how inter-regional brain connectivity is organized and how fine input-output circuit relationships are structured.

Essential revisions:

1) The reviewers felt that further validation of the mouse Cre lines is necessary for the authors to make the major conclusion of the paper.

2) The reviewers pointed to a significant number of errors and omissions throughout the paper, including missing data and organizational problems between the text and figure panels.

3) The reviewers have identified instances in which additional and/or revised statistical analyses are necessary.

*Reviewer #1 (Recommendations for the authors):*

General Comments:

1. The major conclusions of this paper rely on the assumption that Gad1 and Vgat-Cre lines are restrictive to inhibitory neurons and the Ntsr1-Cre line to excitatory neurons. Did the authors confirm the expression of the Gad1-Cre and Vgat-Cre line are indeed restricted to those inhibitory neurons and the Ntsr1-Cre line only to the presumed excitatory neurons? The analyses presented in this paper do not demonstrate overlap or restricted expression between any of the three lines, as they rely on previous expression patterns of genes, not the Cre recombination capability and specificity in the Cre-line(s) itself. The nearly 20% pixel overlap between Ntsr1 and Vgat or Gad1 cells (supplemental figure 1) is also large enough to be a potential confound in the core findings of this paper.

We suggest to test the assumption that each of the Cre-lines is expressed in the intended cell types by co-labelling Cre and Ntsr1/Gad1/Vgat/Vglut2 in your mouse lines using in situ hybridization or co-staining of projections using antibodies for Vglut2 or Vgat. These experiments need to be performed on all three Cre-lines to validate the Cre expression.

2. It is surprising that there are Vgat-positive, Gad1-negative (or vice versa) cells with differential projection patterns (according to Figure 2). The authors need to provide a high-resolution image showing the co-labelling for Vgat and Gad1 neurons demonstrating this apparent incomplete overlap (Figure 1)? Also, were the authors able to find labeled neurons that are Vgat-positive and not Gad1-positive (or vice versa)? Could some of the observed difference be explained by a lower number of infected Gad1 positive neurons than Vgat neurons per injection? Please discuss in the text.

If not, the author may want to change their statements regarding their finding that Vgat and Gad1 populations are not completely overlapping neural subtypes.

3. How did authors differentiate from projection fibers (axons) and projection targets (synapses)? Similarly, how did authors differentiate between fibers and cell bodies?

4. When quantifying "distribution of labeled cells", did the authors observe more unique projections in mice with a lower proportion of labelled cells in the IntA? "Distribution of labeled cells" also may be an ambiguous phrase, as it suggests that the observed proportion of IntA (for example) neurons is labeled according to some map. Perhaps a better title would be "Location of labeled cells." Apologies if I am missing something, in the end it just seems like a leap in the way these data are presented and discussed.

5. It is unclear how input regions were identified, this is especially important given that twice as many Gad1-Cre mice were injected than Ntsr1-Cre mice and that the number of input cells appears a magnitude lower in the Ntsr1-Cre mice. How many rabies-infected cells were labelled for each injection (per mouse and per group)? Does the input region need to be found in all mice from one group, or just one? What is the variability in input region?

And how did the authors control for non-specific labelling? Did they perform any (specifically the rabies tracing) in Cre-negative mice? (methods suggest yes, but results are not show). Please clarify and add the details to the manuscript.

6. Authors should consider including some of the low magnification whole mount images of the supplemental data into the main text figures. They provide a better picture of the data than the cropped images.

Statistics comments:

1. N-number is of neurons is inflated in Figure 3. Statistics should be compared for averages on each mouse, not on total number of cells. The authors seem to perform multiple pairwise comparisons without correcting for multiple comparisons.

Textual comments:

1. Line 113: put abbreviation brackets direction behind "nucleus."

2. Line 114: define "RN."

3. Line 118: define "ZI."

4. Line 119: define "VM, VPM, VAL"

Define all abbreviations in the text first time they are used.

5. Line 116: "a morphological… excitatory neurons." This observation is best left to when figure 5A is discussed, as experimental approach or data describing excitatory neurons has not yet been discussed at this point.

6. Line 124-126: "Vgat neurons generally targeted these regions more strongly than Gad1 neurons." How does this relate to the number of infected neurons in the IntA? This is also a qualitative statement that is not backed by statistical tests.

7. Line 149-155: should authors refer to supplemental figures 2 and 3 here? I would suggest for the authors to order the supplemental figures in the order that they are discussed in the text.

8. Line 179-182: this method may warrant a little more explanation for readers that are not familiar with intersectional viral mapping.

9. Line 182-184: "Injections in wildtype…the nuclei." The authors need to include images of these control experiments in the manuscript.

10. Line 224-225: what tests do these p-values represent?

11. Line 234-235: "we validated…" This is not a validation of your Cre-lines.

12. Line 238-240: "was within the noise of presumptive non-overlapping neurons." Presumed by whom? Almost 20% overlap in signal seems pretty high when you are trying to define cell-type specific projection maps. Please clarify.

13. Line 255-257: "Together, these results…in the brainstem." I think that this conclusion is not warranted based on the data that is shown. See comments 11 and 12 and General comment 1.

14. Line 279: define "nIntA". Why was it chosen to name the putative excitatory neurons based on their Cre-line but the putative inhibitory neurons based on the assumption they are inhibitory (iIntA)? In the end, too many assumptions about the identity of the different neurons were made.

15. Line 289-291: "We currently are unable…neurons project to IO." This is another reason to validate the Cre-lines.

16. Line 291-292: "Exclusive retrograded labeling…ventral brainstem." If this is the case, why can you perform retrograde labeling from IO to iIntA? Are there no iIntA terminals just dorsal to the IO? The supplemental figures do not clarify this.

17. Cite Supplemental Figure 4 in text.

18. Line 333-334: "We conclude it….information broadly." An alternative conclusion is that the majority of Ntsr1-Cre positive cells projects to the RN. I do not think you can conclude that all Ntsr1 cells project everywhere (this is not tested).

19. Line 366-369: Figure 5B only shows proportion of projections. It does not show number of regions targeted or relative strength of these proportions. I think that a qualitative statement can be made that only Ntsr1-Cre neurons project to modulatory regions but the author sshould refrain from making quantitative statements based on proportions of injections that are inherently variable between cell types and injections.

20. Line 398-409: Why are these data not shown?

21. Line 438: how many starter cells did you observe in Gad1 vs Ntsr1 mice?

22. Line 450-451: did the authors test this statistically?

23. Line 441-462: why did the author change the way they referred to Gad1 neurons?

24. Line 460-461: "this patterned… cell types." In previous figures it was shown that iIntA do not heavily project to modulatory regions, whereas nIntA neurons do. Therefore, this pattern does not mirror targeted patterns.

25. Line 535-537: partial difference in projections do not preclude partial overlap in Gad1-Cre and Ntsr1-Cre expression.

Methods comments:

1. What are in the atlas coordinates for the injection sites. Please provide them all.

Figure comments:

1. Please put the abbreviations used in each figure in its figure legend.

2. Please be consistent in the use of "Cre" (capitalized and with an "e") in all figure legends.

3. Label IntA (and other cerebellar nuclei) in larger cross-sections for consistency.

4. Figure 1, use similar schematic of viral injection.

5. Figure 1, please be consistent in labeling of anatomical areas in the figure panels: "Vgat" and "Gad1" above figure panels, and abbreviations for anatomical locations within the figure panel.

6. Figure 1B and 2B, are the Vgat infected cells always more medial than Gad1 infected cells? Is this a problem of injection variability? If so, how do you control for this variability in the different projection maps?

7. Figure 1C-E, boxes in schematics are not same size but figure panels seem similar magnification. Boxes on schematics are a little confusing. It may help to label the anatomical location of interest using color and naming it.

8. Figure 1D, what is the difference between "PSV" and "V" in left images?

9. Figure 2, what does Con/Fon mean?

10. Figure 2, please keep order of Gad1 and Vgat injections consistent with figure 1 for comparison.

11. Figure 3, the statistics for this figure should be rerun, see statistical comments.

12. Figure 4E, see comment 6.

13. Figure 4E, 4th row, there are 3 boxes here.

14. Figure 5 B, please keep color-coding of groups consistently throughout the paper.

15. Figure 6C, provide legend.

16. Figure 6D, consider plotting this data on two images with separate left oriented Y-axis. This figure panel is very unclear.

17. Figure 6F, the leftward and rightwards histograms have been used in other figures to denote ipsi- and contra-lateral projections. Changing this is confusing. It may be beneficial to plot these in the same orientation.

18. Figure 6G, this figure is too small to see and the difference between top panels and bottom panels is unclear.

19. Figure 7, please keep the order in which the Gad1 and Ntsr1 labelled neurons are shown consistent with figure 6 for comparison (keep this order similar with supplemental figure 5 and all figure legends too).

20. Figure 7, retrograde rabies labeled cells are hard too to see, please increase the magnification for input neurons.

21. Figure 7, if I understand correctly, the results for each column are from 4 different injection paradigms/experiments that are also summarized in other figures. It would help the reader if the authors would include schematics for each experiment (include: mouse line, injection side, and type of virus used).

22. In supplemental figure 2, make lines of boxes in inset thicker. Box for Gvi is missing. Box for Hiv is missing. B has no scale bar.

23. In supplemental figure 3, make lines of boxes in inset thicker. Explain scale bars in Figure legend.

24. In supplemental figure 4, make lines of boxes in inset thicker. Label boxes in E-K. Also, B has no scale bar.

25. Table 1, please write out full name of each nucleus in column 1.

*Reviewer #2 (Recommendations for the authors):*

1. For Figure 1, please clarify how much the distinctions in projection distribution could be due to variability in specificity of initial targeting or recombination ability of Vgat-Cre and Gad1-Cre? The conclusions can be strengthened by analyzing the starter distribution of IntA neuron subtypes compared to vestibular or cerebellar cortex like Figure 2C.

The main concern here is that selective targeting of IntA using Cre-lines that recombine in all 3 subnuclei is a potential weakness in methodology since this relies entirely on precision of stereotaxic injection.

2. Questions about similarities or differences between Vgat-Cre and Gad1-Cre recombination or targeting by CN neurons cannot be resolved by in situ hybridization analysis of Vgat and Gad1 in the CN alone (Figure 1A). Perhaps more informative if authors could provide Vgat-Cre; reporter expression with Gad1 in situ and vice versa. Also, it is difficult to see signals in the small panels, and the overlay seems to indicate a lot of heterogeneity. Quantification of Gad1 and Vgat in situ would be informative.

3. Line 177, please describe how labeling of Purkinje cells might affect interpretation. Perhaps clarify that PC labeling might lead to VEST and PBN labeling from Purkinje cell axons directly projecting out of the cerebellum.

4. In Figure 3A, the prediction is that the Gad1-Cre more generally labels Gad1 projections and Gad1IO labels a more restricted subset since using IO retrolabeling. But in 3A, it seems that Gad1IO picked up neurons that were not picked up by Gad1 anterograde labeling, specifically the larger neurons. Text argues perhaps Gad1 labels more interneurons than Gad1IO, but should not exclude Gad1 picking up some of these large neurons (maybe just less). Please explain.

5. Within IntA, what % of vGluT2 neurons does Ntsr1 make up? If Ntsr1 is in almost all, then the result in Figure 4 says almost all nIntA neurons projections are collaterals of RN-projecting nIntA neurons. But if Ntsr1 is in a much smaller subset of IntA vG2 neurons, then this says something different. Analyzing the % of Ntsr1 neurons that are vg2 in the IntA is needed to resolve this.

6. Figure 5A, please provide quantification of the morphology to strengthen the reported differences (PCA analysis?). For IO, wondering if the morphology of CN terminals could fit into the morphological schemes proposed by Vrieler et al., 2019 Brain Structure and Function 224. In other words, do distinct IntA neurons project to specific subtypes of IO neurons.

7. Please show images of the cerebellar cortex in Ntsr1 and Gad1 to show starter cells to address the following questions. For Ntsr1, how much extra-CN recombination does Ntsr1-Cre mediate? For Gad1, I would like to get a sense of how many inhibitory neurons (interneurons and Purkinje cells) are labeled initially. Both are important to determine how to interpret input specifically into CN or whether the authors are looking at input into cell types elsewhere in the cerebellum.

Another way to ask this same question, in Figure 6D, what are the cerebellar cortex Ntsr1 starter cells? Granule cells or maybe even inhibitory neurons? Ntsr1-Cre recombines in excitatory neurons in the CN (Houck and Person 2015), but could more generally label excitatory and inhibitory in the cerebellar cortex. And for Gad1 starter cells in the cerebellar cortex, are they mostly Purkinje cells or do these include Golgi/stellate/basket cells? The concern here is that if the rabies virus receive G and TVA from granule/Purkinje/Golgi/stellate/basket cells, then in addition to CN input, authors are looking at climbing and mossy fibers that project to non-CN cells confounding interpretation.

8. This is important for analysis and interpretation: how does one dissociate the GFP from rabies or GFP from oG? The double GFP strategy is a major concern here.

9. Figure 6B, starter cells are defined as intersection between GFP and mCherry, but in B, there is very little overlap here mostly because red channel is so faint. For instance in Gad1, I can see 2 red cells and 4 green cells but with only 1 yellow cell. I am concerned about the small number of starter cells.

Same issue as Figure 6, in Figure 7B, worried that the input sources represent more than just input sources for CN, but for cell types in the cerebellar cortex especially given high percentage indicated in 6D.

10. For Figure 7A, please provide ways to distinguish axons from dendrites.

*Reviewer #3 (Recommendations for the authors):*

(1a) Interrelationships between the cell types used are not clear. The authors show overlap between Ntsr1+, Gad1+/Vgat+, and VGluT2^+^ neurons (FigS1B). Does this mean that Cre-dependent AAV tracing for Gad1Cre+/VgatCre+ neurons could also label VGluT2^+^ neurons? Does the Ntsr1Cre+ cell type contain inhibitory neurons? If so, how did the authors verify that the axonal projections are actually inhibitory or excitatory? Please discuss the technical limitations.

(1b) Relatedly, it would be helpful to make clear which inhibitory cell types are taken into account and discussed: Gad1-IO, Gad1-collateral, Gad1-GABA-local, Gad1-glycine-local, Gad1-glycine-projection? How did the authors allocate VgatCre+ neurons? What is the overlap between Gad1Cre+ and Vgat1Cre+ neurons (they look different because labeled VgatCre+ neurons are in the medial part of the IntA and labeled Gad1Cre+ neurons are in the central part of the IntA consistently throughout Figures)? Something like Venn diagrams for Gad1Cre+, VgatCre+, and Ntsr1Cre+ neurons, could help.

(2a) I see substantial and consistent injection leak to non-IntA and non-CN regions. How did the authors identify that the connections were really of IntA but not from other regions, for example, Vest? For example, inconsistent with our knowledge are that (1) IntA projections to DCK and PO of IO, (2) IntA projections to several targets (please compare Table 1 with Teune et al., 2000, Prog Brain Res), (3) CrI/PFL Purkinje projections to IntA neurons, which are known to IntP and LN, (4) sparse RN projection to Ntsr1Cre+ IntA neurons, and (5) very dense vestibular projections to Ntsr1Cre+ IntA neurons. Inconsistent within the manuscript is that projection specific tracing revealed more projection targets than non-specific tracing (Table 1). Please discuss these and other relevant technical issues.

(2b) Again, given the technical challenges, the Gad1+/Vgat+ axons labeled in non-IO regions could be explained by injection leak to non-CN regions or to excitatory neurons. On what evidence can the readers be convinced of "inhibitory projections" that are "from IntA"?

3) I see potential issues in viruses used: 1) efficiency for the rabies tracing was significantly different between NtsrCre (yielded ~150 cells) and Gad1Cre neurons (yielded ~4000 cells), 2) rabies tracing didn't identify known input from IO, 3) Cre-independent TVA expression from Cre-dependent AAV has been reported to be a significant problem in the rabies tracing scheme used in this study (Fagat et al., 2016, Cell Rep), and 4) how did the authors control retrograde infection of AAVs? Please discuss the technical limitations.

4) I don't see any evidence for input to VTA.

5) Table 2 is missing.

6) 'long-standing dogma' -- I don't think so. Although it has been 'widely assumed', the existence of collaterals of preolivary neurons has been looked for many times and remains an open question.

7) Difficult to see axonal terminals in photos in general.

8) Cerebellar nuclear outputs are powerfully modulated by their upstream Purkinje cells. In this sense, I think it's worth discussing the organization of PC inputs to inhibitory neurons. Can you discuss more about implication of broader Purkinje innervation of inhibitory IntA neurons? Are these different than the parasagittal modules? Or, does the difference in number/location of labeled PCs for inhibitory vs excitatory DCN neurons simply reflect the technical limitation in rabies tracing?

[Editors' note: further revisions were suggested prior to acceptance, as described below.]

Thank you for resubmitting your work entitled "Widespread inhibitory projections from the cerebellar interposed nucleus" for further consideration by *eLife*. Your revised article has been evaluated by Ronald Calabrese (Senior Editor) and a Reviewing Editor.

The manuscript has been improved but there are some remaining issues that need to be addressed, as outlined below:

Please refer to the specific comments by each Reviewer. In general, the Reviewers have identified areas where the clarity of the figures could be improved. We feel that they will go a long way in helping our readers fully appreciate the data. Also, please address the issues regarding the nomenclature of mouse alleles and how they are presented (e.g. the use of italics). Importantly, all 3 Reviewers have identified a number of important pieces of information that are either missing and/or contain errors.

*Reviewer #1 (Recommendations for the authors):*

In this revised manuscript, Judd et al., have performed essential experiments to validate their Cre-mouse lines. They have incorporated additional mouse lines based on their findings (and previous reviewer comments) and repeated specific experiments in these mouse lines. The observation that inhibitory neurons in the interposed nuclei send widespread projections throughout the brainstem holds and indeed the data present an interesting and important set of findings. However, below are suggestions about the text and figures that will aid the reader in understanding the detailed anatomical data presented in this manuscript.

– The overlap in dots and lines in Figures 1D, 2E, 3D, and 4E make these figures hard to interpret. Some of the projections included in the results may be specific to the injection site outside of the interposed nucleus (off-target viral expression) and this possibility is discussed briefly in the text. In addition, all the regions in which projections are found, even if in just a few viral replicates with the least specificity, are included in the summary of Figure 7B. This should be clearly stated in the text. In addition, we suggest clarifying the panels in 1D, 2E, 3D, and 4E. One option would be to replace these plots with UpSet plots (for example) that show the overlap between the different data sets. Alternatively, the current graphs could be kept and the authors could instead include a ratio of the number of replicates in which the projection was observed. In the end, any way to help the reader more easily appreciate the data would be highly beneficial.

– The authors state "… dorsal to IntA, but nucleocortical terminals that were included in the projection analysis were not located in the same topographical area." Apologies, but I do not understand what you mean by this. What topographical area are you referring to? Please consider rephrasing.

– The authors state "Because injection sites did not differ systematically across…". What do mean? What is your measure that gives you confidence in regard to being systematic?

*Reviewer #2 (Recommendations for the authors):*

In the revised manuscript, the authors have more than sufficiently addressed all my questions and concerns (and those of the other reviewers) about the interpretations of the distinctions of input and output patterns by different Int neuronal subtypes and how the data was generated. The replacement of Ntsr-Cre/Gad1-Cre with vGluT2-Cre/vGAT-Cre mice for monosynaptic rabies tracing clarifies and strengthens the conclusions for the input-output connectivity by excitatory and inhibitory Int neuronal subtypes.

This is a carefully conducted study that describes exciting findings about the anatomical organization of cerebellar interposed anterior nucleus, especially the inhibitory subpopulation. These findings will undoubtedly provide important ground work for future investigation of the functional relevance of distinct Int neuronal subtypes and pathways in dexterous and locomotor movements, and beyond.

I very much enjoyed reading and reviewing this manuscript and congratulate the authors on a well conducted study.

*Reviewer #3 (Recommendations for the authors):*

The manuscript is significantly improved for clarity by the authors' thorough hard work. The existence of ramifying inhibitory projection neurons from Int CN became more convincing. Particularly, analyses on Gad/Vglut immunoreactivity of axonal terminals are helpful.

Now it became clearer that the manuscript adds the widespread ramifying cells to the existing repertoire of inhibitory cerebellar nucleus neurons. The authors establish this cell type by demonstrating the inhibitory signature of projected axons and then place them into context in comparison with well-known types of cells regarding outputs, inputs, and cell morphology. During these efforts, the authors also made clear outputs of a subset of excitatory cells (Ntsr1+ cells) and discovered monosynaptic inputs to the excitatory and 'inhibitory' cells, which themselves are novel and intriguing.

I only have one major comment for improving the clarity of conclusions.

The above efforts made me realize that the current experimental designs do not distinguish or identify the ramifying cells from the widely known IO-only cells (I tentatively call them like this though they could have their own collaterals), which may be important for one to incorporate this cell type into cerebellar theories, although distinction from a subset of excitatory cells is clear. As far as I can see, only sparse and highly restricted labeling in IO in Figure 4-S1D would suggest that IO projections from the ramifying cells are made with a different topographical rule than IO-only cells, which is assuring because these results show a potential distinction between inhibitory cells.

Specifically, in the current manuscript, results for cell morphology, axonal trajectory, and input circuits demonstrated in Figure 5, 6. and 7, which are currently treated as data for the ramifying cells, could simply reflect those of IO-only cells and could barely reflect the ramifying cells. This is because the injection strategies utilized do infect both IO-only and the ramifying cells (To selectively target ramifying cells by avoiding IO-only cells, something like Cre-dependent retrograde infection from RN/PG/TRN in Vgat-Cre mouse would be required). It must be made clear in the Result that these analyses do not distinguish them. This limitation should influence conclusions regarding Figure 5-7 and I recommend modifying them accordingly. Similarly, Line 83-84, Line 376 may also be misleading. It would be clearer to state that what this study identified is "inhibitory projection cell type(s) that (or, at least some of which) collateralizes to IO" rather than "IO projection cell type that also collateralizes to other areas".

---

## [Author Response]

Essential revisions:1) The reviewers felt that further validation of the mouse Cre lines is necessary for the authors to make the major conclusion of the paper.

We have used two independent methods to validate the transgenic Cre driver mouse lines used in the study – in situ hybridization and immunohistochemistry. Of critical importance, we learned that the Gad1-Cre line promiscuously expressed Cre in multiple cell types that project from the interposed nucleus. This promiscuity was observed both in immunohistochemistry, where we found terminal varicosities that expressed Vglut2 or Gad65/67, as well as in somata that expressed Vglut2 and not Vgat, or vice versa, inconsistent with specific label of inhibitory neurons. We could find no report of this promiscuity in the literature, despite the use of this line by other groups. Therefore, as a service to the field we include the description of this validation step in this report, while also entirely removing tracing datasets derived from the Gad1-Cre line from the manuscript. By contrast, we found the Vgat-Cre line to be well controlled with corroborating IHC and ISH data specific to inhibitory cell types. Importantly for the main focus of the manuscript, and as reported in the original submission, we observed that these cells innervated multiple targets outside of IO. The discovery that the Gad1-Cre line was not specific, necessitated replacing Gad1-Cre with Vgat-Cre mice for monosynaptic rabies tracing. We describe details of this discovery, decision, and consequences of this change more thoroughly in the main text of the revised manuscript. All terminal varicosities of Ntsr1-Cre derived projections were Vglut2 immunoreactive, mitigating concerns that these outputs were mixed. However, despite the specificity of immunolabel in terminals, in situ hybridization data in the cerebellar nuclei of Ntsr1-Cre mice revealed a small population of Vgat neurons, putative interneurons, that colocalized with Cre-driven YFP. Therefore, we replaced the Ntsr1-Cre dataset in the monosynaptic rabies experiments with Vglut2-Cre mice. The complete reworking of the monosynaptic rabies datasets in part accounts for the long delay between receiving reviews and this revision. We thank the reviewers for these critical suggestions, which we believe strengthened the main conclusions of the manuscript.

2) The reviewers pointed to a significant number of errors and omissions throughout the paper, including missing data and organizational problems between the text and figure panels.

We have thoroughly revised the manuscript, paying close attention to align text, figures, and figure legends. As described in the point-by-point responses, we have addressed noted errors as well as those not yet noted and are sincerely grateful to the referees for their careful reviews.

3) The reviewers have identified instances in which additional and/or revised statistical analyses are necessary.

We thank the reviewers for these corrections. In the cell size section, where statistical reporting was critiqued, we have (1) expanded the number of mice used, and compared means across animals rather than cells as independent measurements as suggested by the reviewer; and (2) performed statistical tests incorporating adjustments for multiple comparisons (Tukey’s multiple comparisons test), as suggested. The main result – that Vgat-Cre neurons are significantly smaller than Ntsr1-Cre neurons, regardless of direct or projection-specific labeling methods, holds, and is consistent with previously published reports.

Reviewer #1 (Recommendations for the authors):General Comments:1. The major conclusions of this paper rely on the assumption that Gad1 and Vgat-Cre lines are restrictive to inhibitory neurons and the Ntsr1-Cre line to excitatory neurons. Did the authors confirm the expression of the Gad1-Cre and Vgat-Cre line are indeed restricted to those inhibitory neurons and the Ntsr1-Cre line only to the presumed excitatory neurons? The analyses presented in this paper do not demonstrate overlap or restricted expression between any of the three lines, as they rely on previous expression patterns of genes, not the Cre recombination capability and specificity in the Cre-line(s) itself. The nearly 20% pixel overlap between Ntsr1 and Vgat or Gad1 cells (supplemental figure 1) is also large enough to be a potential confound in the core findings of this paper.We suggest to test the assumption that each of the Cre-lines is expressed in the intended cell types by co-labelling Cre and Ntsr1/Gad1/Vgat/Vglut2 in your mouse lines using in situ hybridization or co-staining of projections using antibodies for Vglut2 or Vgat. These experiments need to be performed on all three Cre-lines to validate the Cre expression.

We thank the reviewer for this critical feedback; indeed, the concerns were shared amongst all three the reviewers. As summarized above under Essential Revisions, we have addressed Cre driver line specificity using immunohistochemistry and in situ hybridization in our own tissue, no longer relying on validation through public databases from the Allen Brain Atlas. To summarize changes to the revision that have stemmed from these new experiments: (1) We have added Cre line validation data within each section describing results from the respective Cre lines; (2) In situ hybridization data, along with a variety of hold-out controls, further test for viral specificity; (3) We have removed the Gad1-Cre driver line from the body of the experiments, but retain the characterization of this line as non-specific for the benefit of field; (4) We replace Gad1-Cre driver line with the Vgat-Cre driver line in all sections – direct Cre-dependent tracing, projection-specific tracing, and monosynaptic rabies input tracing; (5) We retain Ntsr1-Cre for output tracing since terminals labeled consistently expressed Vglut2, however we replace this line with Vglut2-Cre for input tracing.

2. It is surprising that there are Vgat-positive, Gad1-negative (or vice versa) cells with differential projection patterns (according to Figure 2). The authors need to provide a high-resolution image showing the co-labelling for Vgat and Gad1 neurons demonstrating this apparent incomplete overlap (Figure 1)? Also, were the authors able to find labeled neurons that are Vgat-positive and not Gad1-positive (or vice versa)? Could some of the observed difference be explained by a lower number of infected Gad1 positive neurons than Vgat neurons per injection? Please discuss in the text.If not, the author may want to change their statements regarding their finding that Vgat and Gad1 populations are not completely overlapping neural subtypes.

The reviewer makes an important point that we think is fully addressed by the observation that the Gad1-Cre line was non-specific in its label of neurons. With this observation in hand, it is perhaps no longer surprising that these projection patterns of Gad1-Cre and Vgat-Cre labeled neurons did not overlap. As noted above, we have removed the Gad1-Cre tracing from the study which we hope fully mitigates the reviewer’s concern.

3. How did authors differentiate from projection fibers (axons) and projection targets (synapses)? Similarly, how did authors differentiate between fibers and cell bodies?

We now describe these criteria more clearly in the text and have added immunohistochemistry to bolster the interpretation of axonal varicosities as synaptic endings (Figures 1, Figure suppl 1-1,1-5 Figure 3, Figure 3 suppl 1). Morphological features, namely swellings along branches of coursing axons helped differentiate axons from synapses. These areas also colocalized markers for neurotransmitter transporters Vglut2 or Vgat. Fibers could be more easily differentiated from cell bodies based on morphological features. While automatic sorters based on fluorescence intensity may sometimes misattribute axons and somata, trained human observers, in my experience, rarely make such attribution errors, which was our approach.

4. When quantifying "distribution of labeled cells", did the authors observe more unique projections in mice with a lower proportion of labelled cells in the IntA? "Distribution of labeled cells" also may be an ambiguous phrase, as it suggests that the observed proportion of IntA (for example) neurons is labeled according to some map. Perhaps a better title would be "Location of labeled cells." Apologies if I am missing something, in the end it just seems like a leap in the way these data are presented and discussed.

This is a great point. We now plot where we saw terminal label as a function of injection site, with color coded plots in Figures 1, 2, 3, and 4. Not surprisingly, the location of terminal label was sensitive to injection site, which we discuss more in depth in the text. We note that even in most restricted injection to anterior interposed in Vgat-Cre mice, we observed axonal varicosities outside of IO. We also thank the reviewer for pointing out that our terminology was confusing. We calculate the fraction of injection site neurons that appear within the cerebellar nuclei and neighboring regions, retitled “Location of injection site” (Figures 1,2,3,4). We now describe more clearly how smaller and smaller injections localized most cleanly in IntA ramify relative to injections that include neurons nearby.

5. It is unclear how input regions were identified, this is especially important given that twice as many Gad1-Cre mice were injected than Ntsr1-Cre mice and that the number of input cells appears a magnitude lower in the Ntsr1-Cre mice.

Vgat-Cre (N=3) and Vglut2-Cre (N=3) replace Gad1-Cre and Ntsr1-Cre, and are N matched. Input regions are simply areas where retrogradely labeled neurons were observed. We clarify this now in the Methods. We think the large difference in input numbers derived from the low numbers of starter neurons in Ntsr1-Cre mice. Vglut2-Cre mice do not show such a stark difference and we now make only one observation about differential innervation of these cells relative to Vgat-Cre, namely that Vgat-Cre cells may be preferentially innervated by neurons of the lateral reticular nucleus.

How many rabies-infected cells were labelled for each injection (per mouse and per group)?

We now quantify the number of starter cells which averaged 156 in Vglut2-Cre and 307 in Vgat-Cre mice. Note that we have modified this experiment by replacing Gad1-Cre mice with Vgat-Cre and included Vglut2-Cre with Ntsr1-Cre to provide a firmer sense of convergence ratios.

Does the input region need to be found in all mice from one group, or just one? What is the variability in input region?

We report all data – so input regions seen in only one mouse are included; such variability would be reflected in the quantification of percent of inputs with error bars in Figure 4 -figure sup1. The raw data are provided in Supplementary file 4.

And how did the authors control for non-specific labelling? Did they perform any (specifically the rabies tracing) in Cre-negative mice? (methods suggest yes, but results are not show). Please clarify and add the details to the manuscript.

We thank the reviewer for pointing out that this was unclear. As alluded to in the methods and now better described and shown, we used rabies in Cre-negative mice as reported for controls. In brief, in Cre-negative mice, rabies (and other recombinase-dependent methods) show little to no label anywhere in the brain (0-8 cells at injection site, and analysis of sections across the brain showed no label anywhere else), bolstering our interpretation of cellular label as “real” signal. Injection of the full complement of viruses – starter viruses plus modified rabies – into wildtype mice resulted in vanishingly little label. This was not comparable to the Cre-dependent tracing which labeled ~60-100s of neurons.

6. Authors should consider including some of the low magnification whole mount images of the supplemental data into the main text figures. They provide a better picture of the data than the cropped images.

This is a great suggestion and indeed very nicely highlights the clear differences in projection patterns observed in the midbrain between Vgat-cre and Ntsr1-cre mice. Low magnification images are shown in Figure 1, 3, and 5. In 5G, side-by-side comparisons are provided for these two genotypes at low magnification, for the reader to appreciate the stark difference in axon tract patterns in the midbrain.

Statistics comments:1. N-number is of neurons is inflated in Figure 3. Statistics should be compared for averages on each mouse, not on total number of cells. The authors seem to perform multiple pairwise comparisons without correcting for multiple comparisons.

We now add per-animal means for Figure 3 cell size data and perform comparisons on means of animals with posthoc tests that account for multiple comparisons.

Textual comments:1. Line 113: put abbreviation brackets direction behind "nucleus."

done

2. Line 114: define "RN."

done

3. Line 118: define "ZI."

done

4. Line 119: define "VM, VPM, VAL"

done

Define all abbreviations in the text first time they are used.

done

5. Line 116: "a morphological… excitatory neurons." This observation is best left to when figure 5A is discussed, as experimental approach or data describing excitatory neurons has not yet been discussed at this point.

Reorganization has fixed this.

6. Line 124-126: "Vgat neurons generally targeted these regions more strongly than Gad1 neurons." How does this relate to the number of infected neurons in the IntA? This is also a qualitative statement that is not backed by statistical tests.

Gad1-Cre data has been removed, and point is taken.

7. Line 149-155: should authors refer to supplemental figures 2 and 3 here? I would suggest for the authors to order the supplemental figures in the order that they are discussed in the text.

Fixed.

8. Line 179-182: this method may warrant a little more explanation for readers that are not familiar with intersectional viral mapping.

Good point. We have added more explanation.

9. Line 182-184: "Injections in wildtype…the nuclei." The authors need to include images of these control experiments in the manuscript.

We now show injections into wildtype mice in Figure S10.

10. Line 224-225: what tests do these p-values represent?

Thank you for catching that this was only in the figure legend; We have re-run the statistics as suggested elsewhere and now report the test used in the main text as well as figure legend (One-way ANOVA; Tukey’s test for multiple comparisons.)

11. Line 234-235: "we validated…" This is not a validation of your Cre-lines.

Please see response to Major point 1 above.

12. Line 238-240: "was within the noise of presumptive non-overlapping neurons." Presumed by whom? Almost 20% overlap in signal seems pretty high when you are trying to define cell-type specific projection maps. Please clarify.

We have replaced analysis of publicly available data from the Allen Brain Atlas (which showed 10 micron brightest point projections which we think accounted for the “overlap” seen), with in situ hybridizations performed with our own mouse lines which show overlap between 1.4-14% of cross-phenotype label in each genotype. As noted above, we removed Gad1-Cre entirely from the study and replaced Ntsr1-Cre with Vglut2-Cre for monosynaptic rabies retrograde tracing.

13. Line 255-257: "Together, these results…in the brainstem." I think that this conclusion is not warranted based on the data that is shown. See comments 11 and 12 and General comment 1.

For points 11-13, we have addressed these limitations with RNAscope in situ hybridization and immunostaining as described more thoroughly in the main critique section. Axonal varicosities from Vgat-Cre mice colocalized with Gad65/67 in multiple targets besides IO. We found this line to be highly specific to Vgat neurons. As a result, we maintain our conclusion that Vgat neurons of interposed and nearby interstitial groups project to multiple targets in the brainstem.

14. Line 279: define "nIntA". Why was it chosen to name the putative excitatory neurons based on their Cre-line but the putative inhibitory neurons based on the assumption they are inhibitory (iIntA)? In the end, too many assumptions about the identity of the different neurons were made.

We have changed the nomenclature to reflect this excellent point. We now use the terms: Int-Vgat, Int-Ntsr1, Int^IO^-Vgat and Int^RN^-Ntsr1 to refer to the various datasets.

15. Line 289-291: "We currently are unable…neurons project to IO." This is another reason to validate the Cre-lines.

Done.

16. Line 291-292: "Exclusive retrograded labeling…ventral brainstem." If this is the case, why can you perform retrograde labeling from IO to iIntA? Are there no iIntA terminals just dorsal to the IO? The supplemental figures do not clarify this.

The targets of these Int^IO^-Vgat neurons are largely restricted to IO, BPN, RN and SPVi. We now draw this distinction out more clearly in the text.

17. Cite Supplemental Figure 4 in text.

Done

18. Line 333-334: "We conclude it….information broadly." An alternative conclusion is that the majority of Ntsr1-Cre positive cells projects to the RN. I do not think you can conclude that all Ntsr1 cells project everywhere (this is not tested).

We clarify that our data suggest that the Ntsr1-Cre neurons that project to RN also project to other locations, but not necessarily everywhere. We take the point that every neuron might not ramify widely and clarify this point in the text.

19. Line 366-369: Figure 5B only shows proportion of projections. It does not show number of regions targeted or relative strength of these proportions. I think that a qualitative statement can be made that only Ntsr1-Cre neurons project to modulatory regions but the author sshould refrain from making quantitative statements based on proportions of injections that are inherently variable between cell types and injections.

We now clarify that projection strength is a semiquantiative metric describing the density of projections, on which we perform statistical tests to support our claims that Int-Ntsr1 targets modulatory regions more than Int-Vgat neurons (See Methods, Projection Quantification section).

20. Line 398-409: Why are these data not shown?

These data, for the retro-Cre injection to RN are now included in a extended figure: Figure 3—figure supplement1.

21. Line 438: how many starter cells did you observe in Gad1 vs Ntsr1 mice?

See response to Point 5

22. Line 450-451: did the authors test this statistically?

We have removed this claim and analysis (referring originally to denser PC projections to Ntsr1-Cre cells).

23. Line 441-462: why did the author change the way they referred to Gad1 neurons?

Nomenclature has been fixed.

24. Line 460-461: "this patterned… cell types." In previous figures it was shown that iIntA do not heavily project to modulatory regions, whereas nIntA neurons do. Therefore, this pattern does not mirror targeted patterns.

With the replacement of the monosynaptic rabies datasets with new Cre driver lines, we find only one statistically significant difference in input patterns between Vgat and Vglut2-Cre cells – those from the lateral reticular nucleus. Thus, we have removed the specific details noted by the reviewer, but addressed the underlying concern with statistical analysis.

25. Line 535-537: partial difference in projections do not preclude partial overlap in Gad1-Cre and Ntsr1-Cre expression.

Indeed. Gad1-Cre has been eliminated and antibody validation has been used to validate Ntsr1-Cre projections.

Methods comments:1. What are in the atlas coordinates for the injection sites. Please provide them all.

All injection coordinates were reported in the Animals section and may therefore have been difficult to find. We have moved them to the Viral injections section.

Figure comments:1. Please put the abbreviations used in each figure in its figure legend.

Done

2. Please be consistent in the use of "Cre" (capitalized and with an "e") in all figure legends.

Done

3. Label IntA (and other cerebellar nuclei) in larger cross-sections for consistency.

We have added a panel to figure 1 indicating boundaries of nuclear subdivisions. Otherwise, as indicated in the text, we use Franklin and Paxinos and Sugihara and Shinoda 2007, to define boundaries.

4. Figure 1, use similar schematic of viral injection.

Done.

5. Figure 1, please be consistent in labeling of anatomical areas in the figure panels: "Vgat" and "Gad1" above figure panels, and abbreviations for anatomical locations within the figure panel.

We have done our best to unify styles of reporting.

6. Figure 1B and 2B, are the Vgat infected cells always more medial than Gad1 infected cells? Is this a problem of injection variability? If so, how do you control for this variability in the different projection maps?

We now plot the locations of the injection sites and have removed Gad1-Cre from the study. We did not observe systematic differences in the location of injection sites across types of tracing.

7. Figure 1C-E, boxes in schematics are not same size but figure panels seem similar magnification. Boxes on schematics are a little confusing. It may help to label the anatomical location of interest using color and naming it.

We have replaced the schematics with low magnification images and boxes now indicate regions with 20X views to the right. We have elected to refer to boxes with lowercase roman numerals. Region abbreviations are provided in all fields of view.

8. Figure 1D, what is the difference between "PSV" and "V" in left images?

We have added lines to denote boundaries between adjacent regions when there might be ambiguities, eg Figure 1Giv

9. Figure 2, what does Con/Fon mean?

This is now defined and described thoroughly in the text:

“Next, to restrict label to genetic- and projection-specific Int neurons (Fenno et al., 2018), we used a two-recombinase-dependent reporter virus (AAV8-hsyn-ConFon-eYFP) injected into Int in conjunction with Flp recombinase retrogradely introduced via IO with AAVretro-EF1a-Flp. The fluorescent reporter will only express in the presence of both Cre and Flp recombinases. This Cre-on Flp-on approach, termed “Con/Fon”, was used to isolate IO-projecting Int-Vgat neurons.”

10. Figure 2, please keep order of Gad1 and Vgat injections consistent with figure 1 for comparison.

Dealt with in revision.

11. Figure 3, the statistics for this figure should be rerun, see statistical comments.

Done.

12. Figure 4E, see comment 6.

Done.

13. Figure 4E, 4th row, there are 3 boxes here.

Fixed

14. Figure 5 B, please keep color-coding of groups consistently throughout the paper.

We have tried to address this.

15. Figure 6C, provide legend.

We have added a key to the panel (now 6D). Good point.

16. Figure 6D, consider plotting this data on two images with separate left oriented Y-axis. This figure panel is very unclear.

We have not included this panel in the revised manuscript.

17. Figure 6F, the leftward and rightwards histograms have been used in other figures to denote ipsi- and contra-lateral projections. Changing this is confusing. It may be beneficial to plot these in the same orientation.

With respect, we have retained the leftward-rightward plotting here but have made the distinction between left and right more transparent and easier to understand with clearer axis labels.

18. Figure 6G, this figure is too small to see and the difference between top panels and bottom panels is unclear.

We have removed this set of panels.

19. Figure 7, please keep the order in which the Gad1 and Ntsr1 labelled neurons are shown consistent with figure 6 for comparison (keep this order similar with supplemental figure 5 and all figure legends too).

We have done our best to improve parallelism.

20. Figure 7, retrograde rabies labeled cells are hard too to see, please increase the magnification for input neurons.

We have done our best to maintain comparable magnification across many panels and balance showing cell groups vs single cells.

21. Figure 7, if I understand correctly, the results for each column are from 4 different injection paradigms/experiments that are also summarized in other figures. It would help the reader if the authors would include schematics for each experiment (include: mouse line, injection side, and type of virus used).

With respect, we think adding the schematics makes this already very large figure more compressed and harder to see the data. We have worked to clarify what is shown better in the legend, and the remaining should orient the reader.

22. In supplemental figure 2, make lines of boxes in inset thicker. Box for Gvi is missing. Box for Hiv is missing. B has no scale bar.

We unfortunately don’t have a great deal of latitude with the boxes because they are added through the microscopy system at the time of capture. We hope the reproduction process maintains the legibility of the boxes. We have fixed remaining issues.

23. In supplemental figure 3, make lines of boxes in inset thicker. Explain scale bars in Figure legend.

Please see response to point 22.

24. In supplemental figure 4, make lines of boxes in inset thicker. Label boxes in E-K. Also, B has no scale bar.

Done, except see note 22.

25. Table 1, please write out full name of each nucleus in column 1.

Done.

Reviewer #2 (Recommendations for the authors):1. For Figure 1, please clarify how much the distinctions in projection distribution could be due to variability in specificity of initial targeting or recombination ability of Vgat-Cre and Gad1-Cre? The conclusions can be strengthened by analyzing the starter distribution of IntA neuron subtypes compared to vestibular or cerebellar cortex like Figure 2C.The main concern here is that selective targeting of IntA using Cre-lines that recombine in all 3 subnuclei is a potential weakness in methodology since this relies entirely on precision of stereotaxic injection.

We thank the reviewer for this keen insight and concern. For all injections, we quantify and plot “location of injection label”. We note in the text where the size of the injection site or specific details of the injection location may account for observed patterns. Of note, injections entirely contained within IntA, and devoid of Purkinje label, were found to project to the pontine nuclei (reticulotegmental and pontine gray), the vestibular nuclei and very sparsely to the parabrachial nucleus. Notably, these injections did not label nucleocortical Vgat terminals. Nucleocortical projections, along with many of the other sensory brainstem label, appeared when injections included neurons of the interstitial nuclei – a loosely organized set of cell groups ventral to the interposed nucleus. We now discuss this observation and draw connections with other previous literature on this topic in both the Results and Discussion.

2. Questions about similarities or differences between Vgat-Cre and Gad1-Cre recombination or targeting by CN neurons cannot be resolved by in situ hybridization analysis of Vgat and Gad1 in the CN alone (Figure 1A). Perhaps more informative if authors could provide Vgat-Cre; reporter expression with Gad1 in situ and vice versa. Also, it is difficult to see signals in the small panels, and the overlay seems to indicate a lot of heterogeneity. Quantification of Gad1 and Vgat in situ would be informative.

We agree with the reviewer have now performed a variety of validation steps that have led to the elimination of Gad1-Cre datasets from the study. We refer the reviewer to our response to Reviewer 1, Point 1 for a thorough description of our validation approaches and findings, prompted by these reviews, including quantitated in situ labeling.

3. Line 177, please describe how labeling of Purkinje cells might affect interpretation. Perhaps clarify that PC labeling might lead to VEST and PBN labeling from Purkinje cell axons directly projecting out of the cerebellum.

We thank the reviewer for raising this point and now include mention of these caveats in the text. Please see our response to Point 1. In addition, we note that in even the most restricted Vgat-Cre IN injection, with no Purkinje neurons labelled, we observed axons entering and ramifying in the medial vestibular nucleus and PB.

4. In Figure 3A, the prediction is that the Gad1-Cre more generally labels Gad1 projections and Gad1IO labels a more restricted subset since using IO retrolabeling. But in 3A, it seems that Gad1IO picked up neurons that were not picked up by Gad1 anterograde labeling, specifically the larger neurons. Text argues perhaps Gad1 labels more interneurons than Gad1IO, but should not exclude Gad1 picking up some of these large neurons (maybe just less). Please explain.

The reviewer is right that this was a puzzling observation. We have eliminated the Gad1-Cre datasets entirely and can only speculate that the peculiar behavior was a consequence of the promiscuous expression of Cre that we observed in this mouse line. We refer the reviewer to more intuitive outcomes of these approaches in Vgat-Cre and Ntsr1-Cre intersectional label.

5. Within IntA, what % of vGluT2 neurons does Ntsr1 make up? If Ntsr1 is in almost all, then the result in Figure 4 says almost all nIntA neurons projections are collaterals of RN-projecting nIntA neurons. But if Ntsr1 is in a much smaller subset of IntA vG2 neurons, then this says something different. Analyzing the % of Ntsr1 neurons that are vg2 in the IntA is needed to resolve this.

We thank the reviewer for this point. We can’t make any claims about the percent of Vlgut2 neurons that Ntsr1 neurons make up because we don’t expect that our virus labels all Ntsr1-Cre cells. We chose to focus on whether the viral labeling approach was specific to Vglut2 or Vgat neurons, and rather than on the fraction of Ntsr1 probe localized with Vglut2 mRNA probes. We therefore back off of claims that collateralization is widespread from individual cells but includes many regions when looked at in aggregate from RN-projecting Ntsr1-Cre cells.

6. Figure 5A, please provide quantification of the morphology to strengthen the reported differences (PCA analysis?).

We appreciate the reviewer’s points here and have addressed the roots of these issues in the following ways. First we better highlight morphological distinctions between projection types in RN and the cerebellar cortex between Vgat-Cre and Ntsr1-Cre projections with new images, but we do not use these as criteria for differentiation, just to point them out qualitatively. We are unaware of PCA methods that could be used to differentiate the morphological features. We note that mossy fiber boutons and the beaded varicosities of the inhibitory nucleocortical projection described by Ankri and observed here are fairly clear distinctions. We have also tried to provide clearer images differentiating the fine structure of the Vgat projection to RN and the differences between the fine caliber axons

For IO, wondering if the morphology of CN terminals could fit into the morphological schemes proposed by Vrieler et al., 2019 Brain Structure and Function 224. In other words, do distinct IntA neurons project to specific subtypes of IO neurons.

This is such an interesting point. However, because we do not fill IO neurons, addressing whether distinct IntA neurons project to specific subtypes of IO neurons is beyond the scope of this study. Importantly, we do not make any claims about this point.

7. Please show images of the cerebellar cortex in Ntsr1 and Gad1 to show starter cells to address the following questions. For Ntsr1, how much extra-CN recombination does Ntsr1-Cre mediate? For Gad1, I would like to get a sense of how many inhibitory neurons (interneurons and Purkinje cells) are labeled initially. Both are important to determine how to interpret input specifically into CN or whether the authors are looking at input into cell types elsewhere in the cerebellum.

We now show and quantify ‘starter’ label for Vglut2-Cre as well as Vgat-Cre mice. As you can appreciate the starter label is located in the cerebellar nuclei and not in the cerebellar cortex. There are no reported direct inputs to Purkinje cells from extracerebellar sources, partially mitigating this concern. However, we only rarely observe Golgi cells labeled with rabies, thus we think that the preponderance of extracerebellar label is from projections to the CN.

Another way to ask this same question, in Figure 6D, what are the cerebellar cortex Ntsr1 starter cells? Granule cells or maybe even inhibitory neurons? Ntsr1-Cre recombines in excitatory neurons in the CN (Houck and Person 2015), but could more generally label excitatory and inhibitory in the cerebellar cortex.

We have replaced Ntsr1-Cre with Vglut2-Cre in monosynaptic rabies tracing. There are not Vlgut2-positive cells in the cerebellar cortex.

And for Gad1 starter cells in the cerebellar cortex, are they mostly Purkinje cells or do these include Golgi/stellate/basket cells? The concern here is that if the rabies virus receive G and TVA from granule/Purkinje/Golgi/stellate/basket cells, then in addition to CN input, authors are looking at climbing and mossy fibers that project to non-CN cells confounding interpretation.

We address this question below Point 7. Beyond modulatory inputs, I am unaware of extracerebellar afferents that target Purkinje cells or MLIs. We noted vanishingly few Golgi cells that were rabies labeled. Please note too, that the differences in inputs between lines was not seen in the new Cre driver lines used for monosynaptic rabies tracing in this revision.

8. This is important for analysis and interpretation: how does one dissociate the GFP from rabies or GFP from oG? The double GFP strategy is a major concern here.

Originally, we count any neuron with both TVA and cell-filling rabies as a starter neuron, noting that it may or may not also express oG. We now include new data with rabies and oG components of monosynaptic labeling expressing different fluorophores, mitigating concern that we did not know if rabies labeled neurons were expressing oG or whether they were labeled via a monosynaptic jump. In the few experiments retaining the original method, oG is easily differentiated from GFP by being histone restricted, thus it labels only a small segment of the nucleus whereas GFP is a cell fill.

9. Figure 6B, starter cells are defined as intersection between GFP and mCherry, but in B, there is very little overlap here mostly because red channel is so faint. For instance in Gad1, I can see 2 red cells and 4 green cells but with only 1 yellow cell. I am concerned about the small number of starter cells.Same issue as Figure 6, in Figure 7B, worried that the input sources represent more than just input sources for CN, but for cell types in the cerebellar cortex especially given high percentage indicated in 6D.

There was very little overlap of red and green channels in the cerebellar cortex, mitigating this concern. We now show better images to appreciate this observation, although the TVA and oG channels is not very photogenic. The large numbers of cells in the cerebellar cortex labeled in from injections (pointed out for 6D) is because Purkinje neurons constitute a major input to the CN.

10. For Figure 7A, please provide ways to distinguish axons from dendrites.

This is a good point and we recognize the reviewers’ concern. We now note in the figure text now that we show terminal fields that contain axonal varicosities. We show colocalization of terminals with markers for neurotransmitter transporters in (Figures 1, Figure suppl 1-1,1-5 Figure 3, Figure 3 suppl 1).

Reviewer #3 (Recommendations for the authors):(1a) Interrelationships between the cell types used are not clear. The authors show overlap between Ntsr1+, Gad1+/Vgat+, and VGluT2^+^ neurons (FigS1B). Does this mean that Cre-dependent AAV tracing for Gad1Cre+/VgatCre+ neurons could also label VGluT2^+^ neurons? Does the Ntsr1Cre+ cell type contain inhibitory neurons? If so, how did the authors verify that the axonal projections are actually inhibitory or excitatory? Please discuss the technical limitations.

We thank the reviewer for pointing out these important caveats. We have added a variety of datasets to address the points raised here. Between in situ hybridization with Cre-dependent reporter viruses and immunolabelling of terminals against either Vglut2 or Gad65/67, we have strengthened the argument that our methods isolate specific subtypes of the cerebellar nuclei. Please see responses within Essential Revisions and Reviewer 1 Point 1 for more details.

(1b) Relatedly, it would be helpful to make clear which inhibitory cell types are taken into account and discussed: Gad1-IO, Gad1-collateral, Gad1-GABA-local, Gad1-glycine-local, Gad1-glycine-projection? How did the authors allocate VgatCre+ neurons? What is the overlap between Gad1Cre+ and Vgat1Cre+ neurons (they look different because labeled VgatCre+ neurons are in the medial part of the IntA and labeled Gad1Cre+ neurons are in the central part of the IntA consistently throughout Figures)? Something like Venn diagrams for Gad1Cre+, VgatCre+, and Ntsr1Cre+ neurons, could help.

As discussed above, we have eliminated the use of the Gad1-Cre line in this study, which may mitigate some concern. We use better terminology to differentiate Vgat neurons labeled from direct injections vs projection-specific label. We also plot and report location of injection sites across all injection types to ensure there are not systematic differences in location of injections across experiments. We do not focus on local inhibitory cell types nor specifically glycinergic cells, though our Vgat terminals colocalized with Gad65/67.

(2a) I see substantial and consistent injection leak to non-IntA and non-CN regions. How did the authors identify that the connections were really of IntA but not from other regions, for example, Vest? For example, inconsistent with our knowledge are that (1) IntA projections to DCK and PO of IO,

We now more directly relate injection site to targets observed, acknowledging the possibility, when appropriate, of projections that could derive from tracer injection location, plotted now in Figures 1,2,3,4. We note when results hold when injection sites are more restricted.

2) IntA projections to several targets (please compare Table 1 with Teune et al., 2000, Prog Brain Res),

We cannot account for all differences, other than to simply report what we observe. We see projections that differ from Teune et al., 2000 even in spectacularly clean injections into IntA (to ipsilateral IO, for example). For other differences, we speculate that label in the interstitial cell groups may account for differences, as this is seen to account for some differentiatinon in PIN targets (T82 in Teune). We now discuss the interstitial cell groups much more thoroughly as the possible source of some of the Vgat projections we observe, as well as the nucleocortical Vgat pathway, which we observed when interstitial cell groups were labeled but not when label was restricted to IntA.

3) CrI/PFL Purkinje projections to IntA neurons, which are known to IntP and LN,

We have de-emphasized IntA specifically and note most starter cells in interposed, with only a minority in LN and interstitial groups. We plot the location of starter cells.

4) sparse RN projection to Ntsr1Cre+ IntA neurons,

We see a sparse RN projection to Vglut2-Cre neurons. This was reported in Beitzel et al., 2017 as well, to Ntsr1-Cre cells.

and (5) very dense vestibular projections to Ntsr1Cre+ IntA neurons.

We now note and cite literature in the text that review vestibular projections that may contradict these findings. (Barmack 2003).

Inconsistent within the manuscript is that projection specific tracing revealed more projection targets than non-specific tracing (Table 1). Please discuss these and other relevant technical issues.

We saw this in the Gad1-Cre tracing (between Int-Gad1 and Int^IO^-Gad1) but not in any other projection specific tracing. In removing the Gad1-Cre driver line from the study we hope the reviewer’s excellent concern is mitigated.

(2b) Again, given the technical challenges, the Gad1+/Vgat+ axons labeled in non-IO regions could be explained by injection leak to non-CN regions or to excitatory neurons. On what evidence can the readers be convinced of "inhibitory projections" that are "from IntA"?

These are very important points. In addition to quantification of injection site label in each dataset, we have amended our claims to reflect the point that our injection sites at times include other cerebellar nuclei. On whole, we now do not emphasize anterior interposed, though we do discuss in sections when we observe terminals correlating with spillage into other nuclei. Of note, we see terminal varicosities in the tegmental reticular nucleus and pontine gray from Vgat-Cre injections even when the tracer is totally limited to IntA, thus we are confident that these projections are not simply a consequence of tracer leakage.

3) I see potential issues in viruses used:1) efficiency for the rabies tracing was significantly different between NtsrCre (yielded ~150 cells) and Gad1Cre neurons (yielded ~4000 cells),

We have replaced both Cre driver lines in the revision. Nevertheless, there is about a 10x difference between number of starter cells in the Vgat-Cre than Vglut2-Cre line. We note this in the text. Given that with the new Cre lines there is very little difference in the pattern of inputs to the two cell classes, we are not concerned that the differences in starter number account for the data.

2) rabies tracing didn't identify known input from IO,

The lack of IO in some specimen is indeed a puzzling feature, and anecdotally is seen by other groups doing similar experiments, as discussed at meetings, possibly due to unknown tropism. With the newer driver lines, we do see IO label in 4/6 specimen. Because we do not see many cells, the input percent is low, thus these known reciprocal projections do not appear in the reciprocal projection map in Figure 7. As has been noted with monosynaptic rabies methods, negative results cannot be interpreted to suggest that specific areas do not project to a given target, but positive results have been validated (Wickersham et al., 2007).

3) Cre-independent TVA expression from Cre-dependent AAV has been reported to be a significant problem in the rabies tracing scheme used in this study (Fagat et al., 2016, Cell Rep),

We thank the reviewer for this concern. We now show control injections in wildtype mice with vanishingly little label suggesting that this caveat is not likely contributing to the main conclusions (Figure S10). We also now include linked TVA-oG construct. Unfortunately we could not find the referenced study to determine if there were other concerns associated with the specific virus used.

and (4) how did the authors control retrograde infection of AAVs? Please discuss the technical limitations.

Retrograde infection of AAVs is easily distinguished by somatic label outside the injection location. We very rarely observed retrogradely labeled neurons in the brainstem and midbrain following our virus injections, suggesting little capacity for terminal label to be misinterpreted. We now note this in the text.

4) I don't see any evidence for input to VTA.

We show this more clearly in Figure2—figure supplement 1, with varicosities amongst TH positive cells in the region of the VTA.

5) Table 2 is missing.

This may have been a typo referencing Supplementary File 3. We thank the reviewer for catching this. We have ensured the tables are all provided.

6) 'long-standing dogma' -- I don't think so. Although it has been 'widely assumed', the existence of collaterals of preolivary neurons has been looked for many times and remains an open question.

We thank the reviewer for this important point. We have removed reference to ‘long-standing dogma’ and replaced it with ‘widely assumed’. In addition, we introduce the question as one that has received empirical support but has nevertheless not been followed up on with cell type specific viral tracing methods (See beginning of Results).

7) Difficult to see axonal terminals in photos in general.

We appreciate the concern. We now include data showing colocalization with Vglut2 or Gad65/67 immunostaining, as described above for R1 and R2 in (Figures 1, Figure 1 figure suppl 1-5 Figure 3, Figure 3 suppl 1).

8) Cerebellar nuclear outputs are powerfully modulated by their upstream Purkinje cells. In this sense, I think it's worth discussing the organization of PC inputs to inhibitory neurons. Can you discuss more about implication of broader Purkinje innervation of inhibitory IntA neurons? Are these different than the parasagittal modules? Or, does the difference in number/location of labeled PCs for inhibitory vs excitatory DCN neurons simply reflect the technical limitation in rabies tracing?

We too were interested in this point, but after replacing the Gad1-Cre datasets with Vgat-Cre, we no longer observed clear evidence of broader innervation of Vgat-Cre cells relative to Vglut2-Cre cells. Thus, we do not have reason to suggest the integrative patterns would differ across cell types.

[Editors' note: further revisions were suggested prior to acceptance, as described below.]

Reviewer #1 (Recommendations for the authors):In this revised manuscript, Judd et al., have performed essential experiments to validate their Cre-mouse lines. They have incorporated additional mouse lines based on their findings (and previous reviewer comments) and repeated specific experiments in these mouse lines. The observation that inhibitory neurons in the interposed nuclei send widespread projections throughout the brainstem holds and indeed the data present an interesting and important set of findings. However, below are suggestions about the text and figures that will aid the reader in understanding the detailed anatomical data presented in this manuscript.– The overlap in dots and lines in Figures 1D, 2E, 3D, and 4E make these figures hard to interpret. Some of the projections included in the results may be specific to the injection site outside of the interposed nucleus (off-target viral expression) and this possibility is discussed briefly in the text. In addition, all the regions in which projections are found, even if in just a few viral replicates with the least specificity, are included in the summary of Figure 7B. This should be clearly stated in the text.

We now point out the inclusion and exclusion criteria for regions listed in Figure 7B – reminding the reader that these include only the strongest and most consistently labeled regions, but nevertheless may reflect projections of interstitial cell groups or other Cb nuclei labeled as described in the earlier figures.

In addition, we suggest clarifying the panels in 1D, 2E, 3D, and 4E. One option would be to replace these plots with UpSet plots (for example) that show the overlap between the different data sets. Alternatively, the current graphs could be kept and the authors could instead include a ratio of the number of replicates in which the projection was observed. In the end, any way to help the reader more easily appreciate the data would be highly beneficial.

We completely agree with the reviewer’s critique of the figure panels that were intended to map the location of projections for each animal. The overlapping scores across replicates made it difficult to see/track the specific pattern of projection targets for individuals, however. Rather than doing an UpSet plot, which would discard information on projection strengths, we elected to offset each specimen slightly from one another and give each specimen a unique symbol. This allows the reader to track each individual while also keeping available the information on injection site precision and projection strengths. We think this edit should fully address the reviewer’s astute observation about the limitations of the previous iteration of these plots.

– The authors state "… dorsal to IntA, but nucleocortical terminals that were included in the projection analysis were not located in the same topographical area." Apologies, but I do not understand what you mean by this. What topographical area are you referring to? Please consider rephrasing.

Reviewer 3 also noted this confusing text and suggested it be removed, so this verbiage is now gone.

– The authors state "Because injection sites did not differ systematically across…". What do mean? What is your measure that gives you confidence in regards to being systematic?

We have replaced the term ‘systematically’ with ‘qualitatively’. The main idea is that if we look at the injection site plots across all the injection types, we do not see that they are categorically different – that is, we could not decipher what the injection scheme was just from looking at the location of the injection site. Nevertheless, we see the Reviewer’s point that the term ‘systematically’ could imply more than we intend and therefore hope the wording change clarifies our point.

Reviewer #2 (Recommendations for the authors):In the revised manuscript, the authors have more than sufficiently addressed all my questions and concerns (and those of the other reviewers) about the interpretations of the distinctions of input and output patterns by different Int neuronal subtypes and how the data was generated. The replacement of Ntsr-Cre/Gad1-Cre with vGluT2-Cre/vGAT-Cre mice for monosynaptic rabies tracing clarifies and strengthens the conclusions for the input-output connectivity by excitatory and inhibitory Int neuronal subtypes.This is a carefully conducted study that describes exciting findings about the anatomical organization of cerebellar interposed anterior nucleus, especially the inhibitory subpopulation. These findings will undoubtedly provide important ground work for future investigation of the functional relevance of distinct Int neuronal subtypes and pathways in dexterous and locomotor movements, and beyond.I very much enjoyed reading and reviewing this manuscript and congratulate the authors on a well conducted study.

We sincerely appreciate these kind words and again express gratitude for the excellent suggestions.

Reviewer #3 (Recommendations for the authors):The manuscript is significantly improved for clarity by the authors' thorough hard work. The existence of ramifying inhibitory projection neurons from Int CN became more convincing. Particularly, analyses on Gad/Vglut immunoreactivity of axonal terminals are helpful.Now it became clearer that the manuscript adds the widespread ramifying cells to the existing repertoire of inhibitory cerebellar nucleus neurons. The authors establish this cell type by demonstrating the inhibitory signature of projected axons and then place them into context in comparison with well-known types of cells regarding outputs, inputs, and cell morphology. During these efforts, the authors also made clear outputs of a subset of excitatory cells (Ntsr1+ cells) and discovered monosynaptic inputs to the excitatory and 'inhibitory' cells, which themselves are novel and intriguing.I only have one major comment for improving the clarity of conclusions.The above efforts made me realize that the current experimental designs do not distinguish or identify the ramifying cells from the widely known IO-only cells (I tentatively call them like this though they could have their own collaterals), which may be important for one to incorporate this cell type into cerebellar theories, although distinction from a subset of excitatory cells is clear. As far as I can see, only sparse and highly restricted labeling in IO in Figure 4-S1D would suggest that IO projections from the ramifying cells are made with a different topographical rule than IO-only cells, which is assuring because these results show a potential distinction between inhibitory cells.Specifically, in the current manuscript, results for cell morphology, axonal trajectory, and input circuits demonstrated in Figure 5, 6. and 7, which are currently treated as data for the ramifying cells, could simply reflect those of IO-only cells and could barely reflect the ramifying cells. This is because the injection strategies utilized do infect both IO-only and the ramifying cells (To selectively target ramifying cells by avoiding IO-only cells, something like Cre-dependent retrograde infection from RN/PG/TRN in Vgat-Cre mouse would be required). It must be made clear in the Result that these analyses do not distinguish them. This limitation should influence conclusions regarding Figure 5-7 and I recommend modifying them accordingly. Similarly, Line 83-84, Line 376 may also be misleading. It would be clearer to state that what this study identified is "inhibitory projection cell type(s) that (or, at least some of which) collateralizes to IO" rather than "IO projection cell type that also collateralizes to other areas".

We have now added modifiers and text in a variety of places that states more clearly that our data indicate that at least some IO-projecting neurons collateralize and that we cannot distinguish neurons that solely project to IO, and that our data do not preclude their existence. Eg: Lines 84; 177-178; 379; 444-445.

Also, upon reflecting on this point, we have elected to tone down the title a touch so as not to unintentionally mislead the reader that all inhibitory projections are widespread. To this point, we have replaced ‘widespread’ with ‘diverse’. We hope this minor edit also addresses the spirit of the concern raised here.